# Distribution of heavy metal resistance elements in Canadian *Salmonella* 4,[5],12:i:- populations and association with the monophasic genotypes and phenotype

**Clifford G. Clark**[1]*, **Chrystal Landgraff**[1], **James Robertson**[2], **Frank Pollari**[3],
**Stephen Parker**[3], **Celine Nadon**[1,4], **Victor P. J. Gannon**[5], **Roger Johnson**[2], **John Nash**[2]

**1** National Microbiology Laboratory, Public Health Agency of Canada, Winnipeg, Manitoba, Canada,
**2** Division of Enteric Diseases, National Microbiology Laboratory, Public Health Agency of Canada, Guelph,
Ontario, Canada, **3** FoodNet Canada, Public Health Agency of Canada, Guelph, Ontario, Canada,
**4** PulseNet Canada, Public Health Agency of Canada, Winnipeg, Manitoba, Canada, **5** Division of Enteric
Diseases, National Microbiology Laboratory, Public Health Agency of Canada, Lethbridge, Canada

* clifford.clark@canada.ca

doi.org/10.1371/journal.pone.0236436

Department of Health, UNITED STATES

**Data Availability Statement:** The datasets
generated or analyzed in the current study are
available in the National Center for Biotechnology

## Abstract

*Salmonella* 4,[5],12:i:- are monophasic *S.* Typhimurium variants incapable of producing the
second-phase flagellar antigen. They have emerged since the mid-1990s to become one of
the most prevalent *Salmonella* serotypes causing human disease world-wide. Multiple
genetic events associated with different genetic elements can result in the monophasic phe-
notype. Several jurisdictions have reported the emergence of a *Salmonella* 4,[5],12:i:- clone
with SGI-4 and a genetic element (MREL) encoding a mercury resistance operon and antibi-
otic resistance loci that disrupts the second phase antigen region near the *iroB* locus in the
*Salmonella* genome. We have sequenced 810 human and animal Canadian *Salmonella* 4,
[5],12:i:- isolates and determined that isolates with SGI-4 and the mercury resistance ele-
ment (MREL; also known as RR1&RR2) constitute several global clades containing various
proportions of Canadian, US, and European isolates. Detailed analysis of the data provides
a clearer picture of how these heavy metal elements interact with bacteria within the *Salmo-
nella* population to produce the monophasic phenotype. Insertion of the MREL near *iroB* is
associated with several deletions and rearrangements of the adjacent *flaAB hin* region,
which may be useful for defining human case clusters that could represent outbreaks. Plas-
mids carrying genes encoding silver, copper, mercury, and antimicrobial resistance appear
to be derived from IS26 mediated acquisition of these genes from genomes carrying SGI-4
and the MREL. Animal isolates with the mercury and As/Cu/Ag resistance elements are
strongly associated with porcine sources in Canada as has been shown previously for other
jurisdictions. The data acquired in these investigations, as well as from the extensive litera-
ture on the subject, may aid source attribution in outbreaks of the organism and interven-
tions to decrease the prevalence of this clone and reduce its impact on human disease.

Information (NCBI) repository under BioProject PRJNA543337. Biosample numbers for each isolate can be found in S1 Spreadsheet.

**Funding:** Government of Canada Genomics Research and Development Initiative (GRDI) - Phase VI - CGC, RJ Government of Canada A-base program funding - CGC, RJ, JN, JR, FP.

**Competing interests:** The authors have declared that no competing interests exist.

## Introduction

*Salmonella enterica* subsp. *enterica* serovar 4,[5],12:i:- is an emerging *S.* Typhimurium monophasic variant that has dramatically increased in prevalence world-wide since the mid-1990s [1–3]. During this same period the prevalence of *S.* Typhimurium has declined and that of *S.* Enteritidis has dramatically increased in Canada [4]. For the last seven years, *Salmonella* 4,[5],12:i:- has been the 4th or 5th most prevalent *Salmonella* serotype, or higher, in Canada [4,5], Europe [1], the U.S. [6], and other countries [7–10].

The *S.* Typhimurium second-phase flagellar antigen locus includes *fljB* (second phase flagellin gene), *fljA* (repressor of transcription of the first phase flagellin gene), and *hin* (DNA invertase for a segment of DNA containing the promoter for transcription of the *fljA* and *fljB* genes) [2,3]. Isolates belonging to *Salmonella* 4,[5],12:i:- clones were first characterized by PCR for the presence of the *fljAB hin* genes constituting the second-phase flagellar antigen locus [1,11,12], as well as adjacent genes and deletions [13]. Characterization of monophasic isolates from chickens and pigs identified a number of deletions associated with *fljA*, *fljB*, and *hin*, including partial deletions and mutations of *hin* and *fljB* [14,15]. The earlier work [14] also inferred the existence of smaller deletions and point mutations not observed directly. Several deletions of different length with different endpoints have been identified [16], as well as DNA rearrangements [17] and point mutations within the second phase antigen locus [18,19]. Deletions and rearrangements appear to generate a stable genotype/phenotype, but both the point mutations in *fljA*, *fljB* and *hin* and the IS26 elements disrupting different intergenic regions of the *fljAB hin iroB* locus [17] may be reversible and result in an intermittent or inconsistent phenotype.

Different clones of *Salmonella* 4,[5],12:i:- have emerged in different countries at different times [2,3,5,7,19–21] but have generally been assigned to three dominant types, the Spanish clone, the European clone, and the U.S./American clone [2]. Isolates with deletions at the *fljAB hin* locus different than these three clones were detected in monophasic isolates in China [22]. The designation of clones was historically not based on the lesions at the *fljAB hin* second phase flagellin locus but on other characteristics of these bacteria, frequently characterization using phenotypic and molecular typing methods [23–25].

Characterization by microarray identified characteristic features of the Spanish clone, including a deletion of the allantoin operon, absence of both the FELS-1 and FELS-2 prophages, loss of two genes within the Gifsy-1 prophage, and a 16 gene deletion that included *fljAB hin* and only part of *iroB*, leaving an IS26 element adjacent to the remaining part of the *iroB* genes [26,27]. One group of phage type U302 isolates belonging to the Spanish clone was determined to be multi-drug-resistant and carry four different deletions of the second phase antigen region, all beginning within STM2758 but leaving a partial or complete *iroB*, deleting *iroB* altogether but leaving *iroC*, or deleting this region plus all genes downstream to *emrA* [13]. While clonal in some respects, the "Spanish clone" exhibited a great deal of variability in the lesion at the *fljAB hin* locus.

There has been expansion of an epidemic clone in the UK defined by WGS that is resistant to arsenate, silver, and copper (As/Cu/Ag) due to the presence of SGI-4 and resistant to mercury and multiple antibiotics subsequent to integration of a transmissible element [16]. This clone was found in the US as well, suggesting world-wide spread from a single source [15]. SGI-4 is an Integrative Conjugative Element (ICE) capable of transmission between strains [28]. Increased resistance to antibiotics has been proposed as a means by which this clone and some others survive and come to predominate. The use of heavy metals may also provide a selective mechanism that partly explains the expansion of mercury- and As/Cu/Ag- resistant clone(s); carriage of antimicrobial resistance genes on the same chromosomal element as the

mercury resistance genes may also be partly responsible [15]. There is a strong association of the heavy metal resistant clone(s) with pork production attributed to the use of heavy metal growth promoters in pork production, especially after the European ban on use of antimicrobials for that purpose [29]. Heavy metal resistance could allow bacteria to survive macrophage killing mediated by release of large amounts of copper and zinc into phagosomes, and resistance to heavy metals encoded by As/Cu/Ag genes was shown experimentally to confer resistance to copper toxicity affecting motility [29].

Chromosomal elements encoding resistance to mercury have been described in *Salmonella* 4,[5],12:i:- isolates [15–17,30,31] and these are associated with multiple genetic polymorphisms involving the second phase flagellar antigen locus that result in the monophasic phenotype [15,16,30]. The resistance regions RR1 and RR2 described by Lucarelli and colleagues [31] are identical to what we have named here as the <u>m</u>ercury <u>r</u>esistance <u>el</u>ement (MREL) in order to assess it as a discrete chromosomal island. At least three different configurations of the element were present within different isolates, and the genetic lesions (deletions or rearrangements) created by insertion of this element near *iroB* were different. These deletions appeared to be stable [15,16], suggesting they may have arisen from different events though they were found within an expanding clone within the UK. A logical hypothesis resulting from these findings is that characterization of the specific deletions associated with the MREL may be useful for surveillance and cluster detection of isolates carrying them. The deletions at *fljAB hin iroB* may be the result of activity of the composite Tn21-like transposable element carrying the mercury operon [16] or IS26 elements [30]. Because the *fljAB hin* locus has a % G + C content of 41%, much lower than the average for *S.* Typhimurium [32], it has been proposed that this region may be a hotspot for recombination [30,32].

Both the MREL and SGI-4 are frequently detected in *Salmonella* 4,[5],12:i:- isolates. However in one study the two elements assorted independently and were not significantly associated with each other in an analysis of isolates from food animals, and copper/silver resistance was detected more frequently than mercury resistance [29]. Isolates comprising the European monophasic clone carried the MREL and SGI-4 in the chromosome, while isolates from the Spanish clone carried these elements in plasmids [33]. These plasmids may be similar to plasmid pU302L from the U.S. phage type U302 *S.* Typhimurium isolate G8430 [34] or the *E. coli* plasmid pO111_1 [16,30], which both carry the mercury resistance operon. The existence of plasmids in the paper by Chen et al. [34] was determined by conjugation; it is not clear whether these plasmids were also capable of chromosomal integration and excision.

In our current investigations we are exploring the DNA sequences of Canadian *Salmonella* 4,[5],12:i:- isolates using closed genome sequences as references for analysis to: 1) infer a population structure for the isolates; 2) compare genome sequences of Canadian isolates to those from isolates world-wide; 3) identify the full complement of genetic lesions responsible for the monophasic phenotype in the isolates selected; 4) develop a database of genome sequences for use in laboratory and epidemiological surveillance activities as well as cluster identification, outbreak characterization, and association with source. All analyses presented here focus on the characterization of isolates that carry elements encoding heavy metal resistance and associated antimicrobial resistance (AMR) genes or genetic elements. The data show that Canadian isolates carrying the MREL and SGI-4 are part of the world-wide clone described by Elnekave et al. [15].

## Materials and methods

### Isolates, growth conditions, DNA preparation, and sequencing

Clinical isolates were obtained from Canadian provincial public health laboratories and permission for use was granted by the Canadian PulseNet Steering Committee and provided to

the National Microbiology Laboratory (NML) in Winnipeg, Canada. Additional human iso-lates and all non-human isolates were provided by FoodNet Canada to the NML facility in Guelph, Canada. Isolate numbers were replaced with randomly generated numbers for publi-cation. The isolates sequenced for this project do not constitute a random sample from avail-able collections. Preferred isolates 1) were part of an outbreak of human disease that was investigated epidemiologically; 2) were obtained within 60 day time periods before the first case and after the last case in the outbreak; 3) belonged to a PFGE cluster of four or more; 4) had PFGE results; 5) were obtained from geographically widespread areas; 6) could help define the breadth of the population; 7) were from as many non-human sources as possible; 8) had metadata available; 9) were readily available within existing collections at the Winnipeg or Guelph laboratories or could be readily obtained. The selection of isolates was not unbiased. For example, isolates from the 1008ST399MP outbreak and overlapping clusters with PFGE pattern STXAI.0008 were highly over-represented within the population assessed. Permission to use isolates was obtained by the authorities responsible, e.g. provincial public health labora-tories. Selected metadata for isolates sequenced for this study may be found in S1 Spreadsheet.

Isolates were stored in 19% skim milk or in peptone with 20% glycerol at -80˚C and were routinely grown on Nutrient Agar (NA) containing 1.5% NaCl or in lysogeny broth (LB) at 37˚C for DNA isolation. PFGE was done by PulseNet Canada using the PulseNet standard method [35] and banding patterns were analyzed using Bionumerics (version number depend-ing on the year). Isolation was carried out either with Epicentre Metagenomic DNA Isolation Kit for Water (Illumina) or the DNeasy® 96 Blood & Tissue Isolation Kit (Qiagen). Isolated DNA was sent to the Genomics Core facility at the NML for quantitation and dilution, and sample libraries were prepared using a MiSeq Nextera® XT DNA library preparation kit (Illu-mina). NextSeq has previously been used successfully for WGS of *Salmonella* 4,[5],12:i:- iso-lates [36]. Whole genome sequencing was therefore performed by 150 bp paired-end read sequencing on the Illumina NextSeq platform using NextSeq 500/550 Mid Output kits and sequence reads were deposited in the appropriate IRIDA Platform Beta Release databases (ngs-archive.corefacility.ca/irida) [37].

## DNA sequence quality, assembly, and annotation

Sequence reads were assembled into contigs using the SPAdes assembler (v3.0) [38]. Contigs smaller than 1 kb and with average genome coverage less than 15× were filtered and removed from the analysis. Draft genome sequences were annotated using PROKKA [39]. Sequence and assembly quality were assessed using QUAST in IRIDA and the quality tools in Bionu-merics v. 7.6.2 (Applied Maths). Quality measures from BioNumerics for assembled genomes can be found in S2 Spreadsheet.

## DNA sequence analysis and bioinformatics

The initial analysis involved manual inspection of all.gbk files by searching for the presence of *iroB*, *hin*, *fljB*, and *fljA* genes using gene or protein descriptions or short unique peptides for each locus. In the one instance where iroB was not detected we systematically searched upstream and downstream until the nature of the deletion was clear. Similarly, annotated mer-cury resistance genes associated with the MREL were detected using the search term "mer-curi"; once the mercury resistance operon was determined the identity and sequence of the surrounding genes was characterized. A similar process was used to detect and characterize SGI-4.

SNP analysis was done using the SNVPhyl program implemented as a Galaxy workflow [40] and Maximum Likelihood trees were constructed within the same workflow. Details of

the workflow parameters and outputs are summarized in S1 File. Dendrograms were visualized and annotated in FigTree v1.4.4 [41]. WGS data was also used to create wgMLST dendrograms in BioNumerics. GView server (https://server.gview.ca) was used for data visualizations in the form of displaying genome features or BLAST-based pangenome analysis using closed genomes [42,43] as references. GView [44] is an application that uses.gbk or.fasta sequence files for: genome mapping and visualization of individual genomes, comparison of more than one genome to determine core and accessory genomes, or comparison of multiple genomes against a reference closed genome to determine the presence/absence and % identity of genes and proteins in BLAST-based analysis. Verification of features occurring in these visualizations was checked manually in.gbk files of the relevant genomes. Figures were prepared using Adobe Illustrator CC 2018.

Incompatibility types of plasmids and the MREL were determined using PlasmidFinder 2.1 at the Center for Genomic Epidemiology (cge.cbs.dtu.dk/services/PlasmidFinder).

## Results and discussion

Draft whole genome sequences were obtained for 811 *Salmonella* 4,[5],12:i:- isolates. Of these 6 were initially closed for use as references for further analysis [42], with another 40 closed later [43]. Quality metrics for all assemblies were obtained using Bionumerics (S2 Spreadsheet); the data include results from a comparison of 32 isolates sequenced from the same library using both MiSeq reagents and instruments and NextSeq reagents and instruments. Results were comparable. Though the quality was generally somewhat better with MiSeq sequencing, it was acceptable with NextSeq sequencing as well.

### The MREL is associated with deletions of *fljA*, *fljB*, and/or *hin* through mobile element integration and imprecise excision

Both the MREL and SGI-4 were detected in similar genomic locations in most *Salmonella* 4,[5],12:i:- isolates with closed genomes; the two elements are most frequently separated by approximately 1700 kbp of intervening genomic sequence (Fig 1). The MREL was detected near *iroB* or adjacent genes (depending on the deletion resulting from its insertion) in closed and finished reference genomes as well as all draft genomes in which the MREL and *iroB* were on the same contig (Fig 2). It is composed of an assemblage of operons adjacent to transposon or integrase genes that appeared to define at least four mobilizable elements (Fig 2). Heterogeneity was evident from the loss of transposases, the *tnpR* and β-lactamase genes, the mercury resistance mobilizable element, the tetracycline resistance mobilizable element (isolate PNCS010567, Fig 2), and a fourth element including genes from the Tn10 transposase through the *dcm* gene encoding DNA cytosine methyltransferase (see PNCS014863, Fig 2). Genes encoding *repC* and *repA* also appeared to be capable of excision from the MREL in isolate PNCS015046 (Fig 2). In isolate PNCS014863 there was a deletion of the 3´ region of the *repC* gene as well as the 3´ region of a gene encoding a DDE domain protein. The *hin* gene was either truncated or completely absent in the isolates shown, and *fljAB* genes were either intact, contained point mutations, or were missing completely (Figs 2 and 3). An integrase containing transposase IS26 was the first upstream gene of the inserted element and either lay adjacent to the *iroB* gene (most genomes shown) or to the gene remaining after a longer deletion that removed *iroB-iroN* (PNCS014863, Fig 3).

There was extensive heterogeneity evident with the MREL (Figs 2–4). Heterogeneity was detected earlier in the element described by Boland et al. [16] (see Fig 4), though the effects of insertion of the MREL in their isolate KJ999732 caused local genome rearrangement and did not result in the loss of *fljAB hin*. The substantial heterogeneity of the MREL makes it difficult

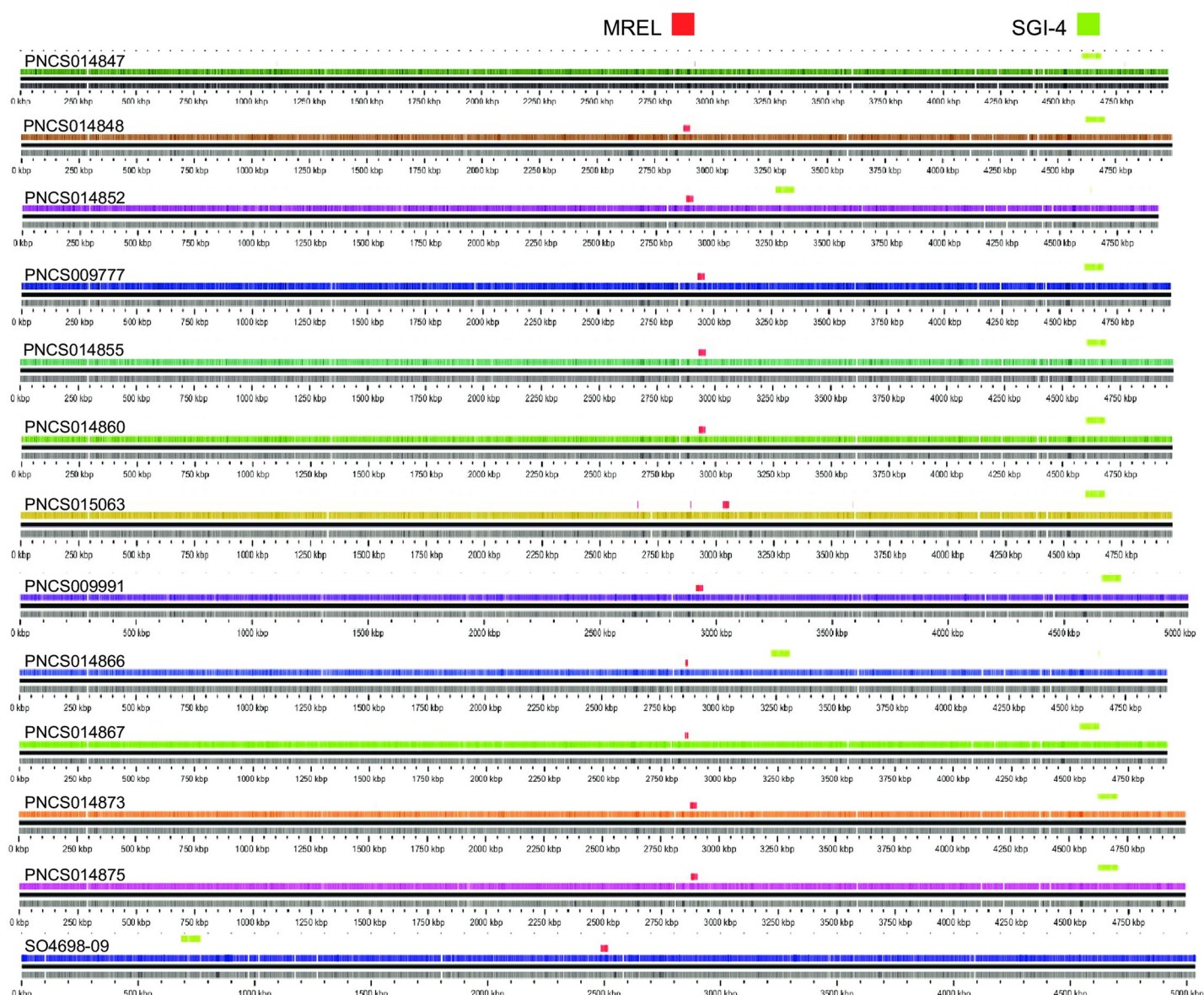

**Fig 1. Location of the mercury resistance element MREL and SGI-4 within closed genome sequences.** The comparison was done using GView Server as four separate "BLAST atlas" analyses and the resulting image files were assembled and annotated in Adobe Illustrator CS6.

to assign a single marker diagnostic for the element. Most of the heterogeneous regions were adjacent to transposases, suggesting the transposases were responsible for the genetic malleability of the MREL, though the frequency of change in closely related or clonal populations was not addressed in this work. It is possible that the MREL changes too quickly to be useful for epidemiological investigations. However, variability of deletions and mutations the adjacent *fljAB hin* genes resulting from MREL integration may be very useful for validating case clusters and investigating outbreaks. Two isolates (PNCS014854, PNCS015054) without the MREL were included in Fig 3 for comparison with MREL-containing isolates. Both isolates carried an intact *hin* and had a deletion from *fljB* through the entire FELS-2 prophage, demonstrating variability at the *fljAB hin* region that is not associated with the MREL. These isolates will be examined further in another paper. Interestingly, *S.* Typhimurium strain DA3482

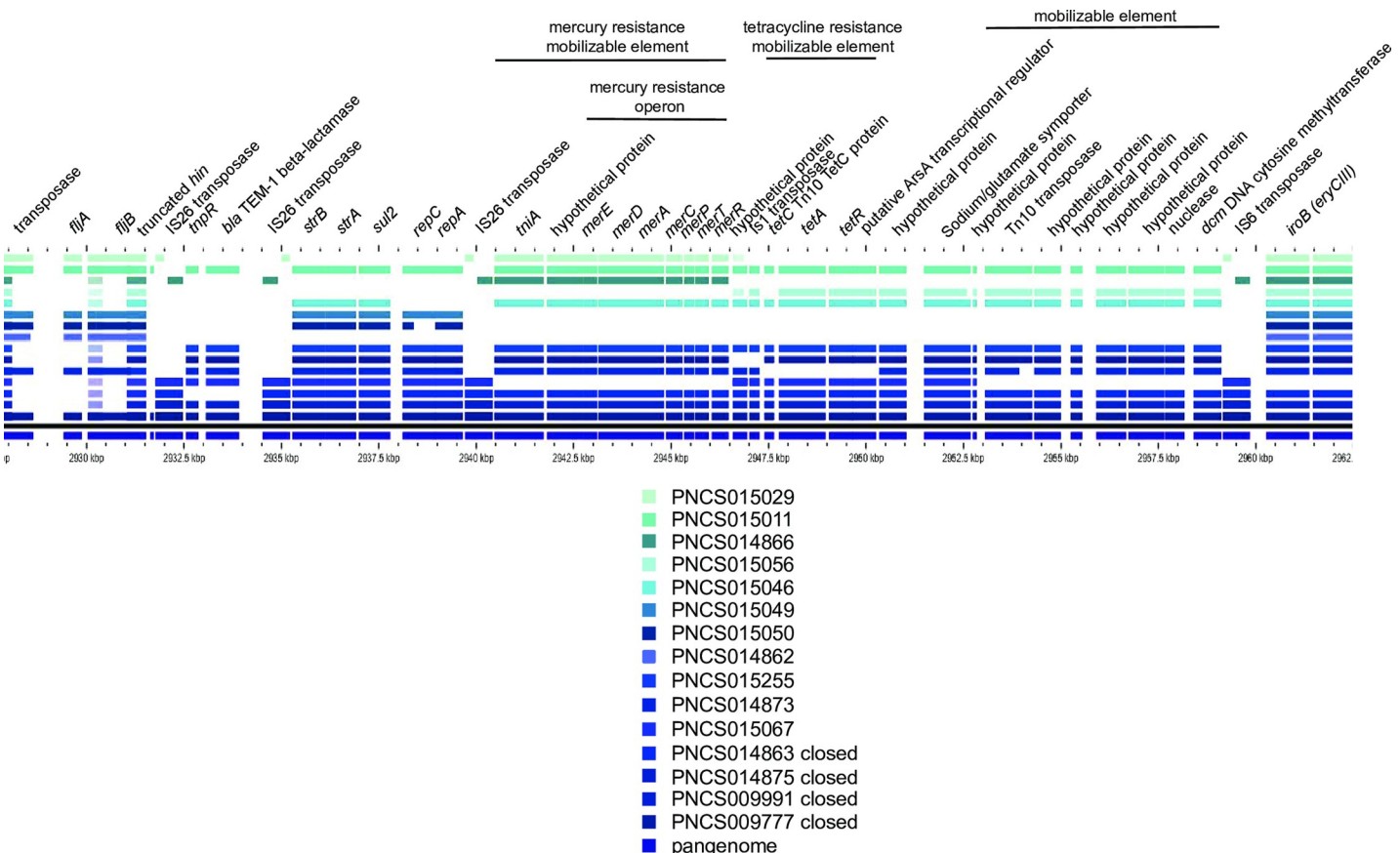

**Fig 2. Identification and structure of mobilizable elements showing heterogeneity within the MREL.** Both draft and closed genomes were used to create the diagram. The comparison was done using GView Server with the "pangenome analysis" settings and the figure was annotated in Adobe Illustrator CS6. This analysis uses BLAST searches to align genes to a reference genome, in this case isolate PNCS009777. Note that the *tetA* designation is used throughout to describe the gene encoding the class B tetracycline determinant tet(B). Not all loci defined by Lucarelli and colleagues [31] are shown.

(GenBank accession no. CP029567.1) carried an insertion between *iroB* and an intact *hin* gene derived from the first few genes of the MREL. Three other strains in GenBank (DA34837, accession no. CP029568.1; 81741, accession no. CP019442; TW-Stm6, accession no. CP019649) were *Salmonella* 4,[5],12:i:- with partial or complete MREL and complete SGI-4.

Previous figures summarized the MREL content of the isolates, but did not investigate synteny of genes with MRELs from different isolates. While four genomes (PNCS009777, PNCS009991, PNCS014875, and SO4698-09) showed almost complete synteny except for IS elements, the MREL-containing region previously described by Boland et al. [17] and our isolate PNCS014863 both showed rearrangements and deletions consistent with the relatively large number of IS elements associated with the region (Fig 4). Both isolate PNCS014863 and the Boland et al. [17] sequence with Accession no KJ999732 contained chromosomal or prophage genes from regions of the chromosome adjacent to *fljAB hin*, suggesting that genome rearrangement is not an uncommon event. We think that these rearrangements may be tied to integration of the MREL into the chromosome. The presence of the truncated *sgrR* (STM2693 in *S*. Typhimurium LT2) in isolate PNCS014863 supports this hypothesis; however, loss of the mercury resistance cassette could have occurred upon integration of the MREL or later. It is clear that this region is quite dynamic (see S1 Fig). Searching the PNCS009777 MREL in

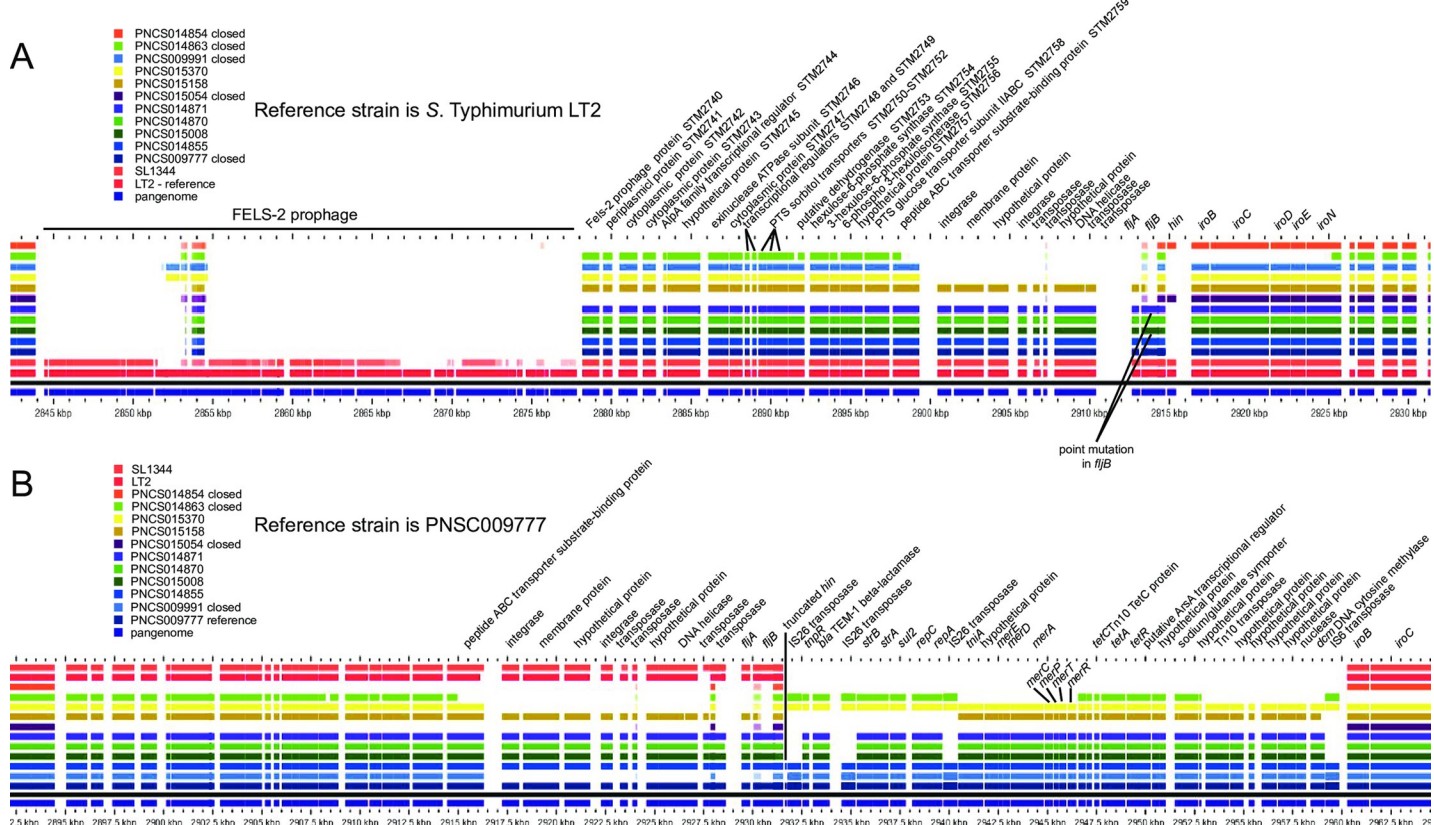

**Fig 3. *Salmonella* 4,[5],12:i:- isolates showing diversity of changes in the genome at the second-phase antigen locus.** A. *S.* Typhimurium LT2 was used as the reference to produce the alignments. B. Reference isolate PNCS009777 was used as the reference. The analysis was done using GView Server using "Pangenome Analysis" settings.

PlasmidFinder 2.0 determined that IncQ incompatibility was encoded in *repA* sequences within the MREL.

The presence of substantial degradation of the MREL in closed genomes (S1 Fig; especially PNCS014866, PNCS0148876, and PNCS014867) makes it difficult to determine whether the MREL had been inserted into the genome and was subsequently lost or whether some portion of the MREL had inserted independently into genomes, perhaps mediated by IS26 transposons (see, for example, PNCS014866). It is clear from these data that MREL structure is highly plastic and that much of the changes seen appear to be associated with regions bounded by IS26 elements. The region containing a hypothetical protein, beta-lactamase TEM-1, and an IS91 transposase pseudogene (green) is inverted between isolates PNCS01114852 and PNCS01 14855, while the region containing *strA*, *strB*, *sul2*, *repC*, and *repA* (red) is inverted by PNCS0114848 and PNCS0114852 groups of isolates. The entire region between *dcm* and *strB* (red), which is bounded by IS26 elements, is inverted in PNCS014861 versus other genomes assessed. Finally, a complex rearrangement is found in PNCS0114867 with one MREL region containing TEM1 beta-lactamase (red), rearranged chromosomal genes, and a second distal MREL region containing *tetAR* and a number of other MREL genes (red, S1 Fig). IS26 elements are distributed throughout and may be associated with the mechanism of rearrangement.

In aggregate, the findings are consistent with the occurrence of repeated integration into *S.* Typhimurium chromosomes at or near the *fljAB hin* locus to create the different deletions

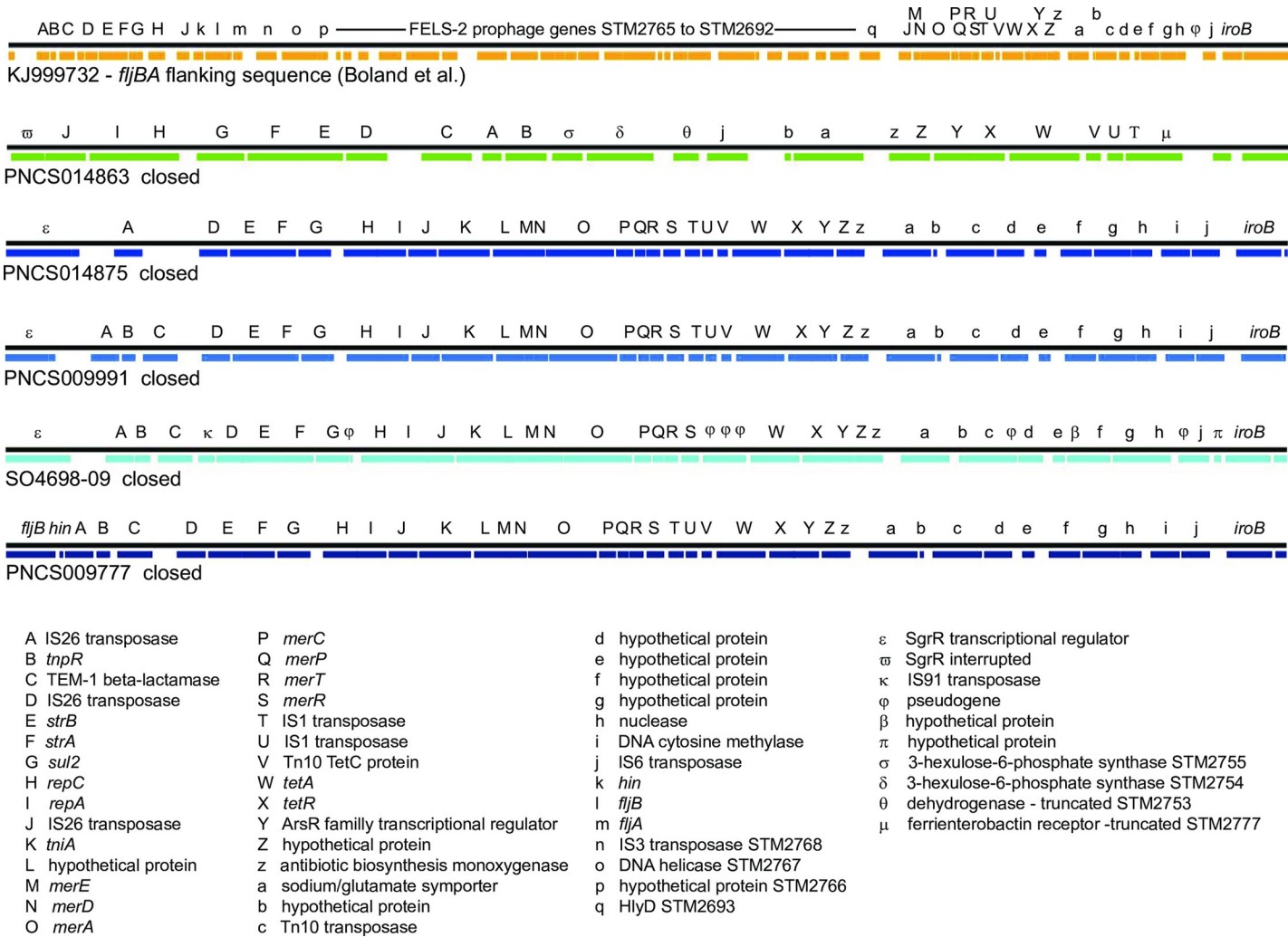

**Fig 4. Synteny and variability of the MREL in *Salmonella* 4,[5],12:i:- closed genomes.** In addition to isolates sequenced in this study, strain SO4698-09 and the rearranged locus described by Boland et al. [16] were included. Analysis in GView Server used the "Display genome features" settings. The resulting image files were assembled and annotated in Adobe Illustrator CS6.

detected in both this work and in Petrovska et al. [16], though clonal descent is also quite apparent in dendrograms and MSTs.

Recent work demonstrated that the unaided deletion of *fljBA* occurs at very low frequencies [45]. The IS26 element alone is capable of inserting near *iroB* [17], consistent with the observation that the region has a different % G + C content than the genome average and could be a hotspot for recombination [30,32]. However, it is not yet clear whether the IS26 element alone is responsible for the deletion of adjacent regions of DNA or whether it is a remnant left from the integration and later excision of a plasmid or other element containing the MREL. Attempts to mobilize the MREL from donor strains to recipients were unsuccessful [46].

Characterization of the specific genetic lesion resulting from integration of the MREL with accompanying changes to the *fljAB hin iroB* region and adjacent chromosomal regions (see S1 Fig) provides—strain-specific data in addition to SNP and wgMLST analysis. Our current assumption is that this insertion event leads to a genetic configuration (deletion) that is stable, at least on the order of epidemiologically meaningful case clusters or outbreaks. Six different

genetic lesions responsible for the monophasic phenotype associated with the MREL have been characterized in this work using both *S.* Typhimurium (Fig 3A) and closed *Salmonella* 4,[5],12:i:- strain PNCS009777 whole genome sequence as references for BLAST searches and pangenome data visualization using GView server (Fig 3B and S1 Fig). These are as follows: 1) deletion in isolates PNCS009777 and PNCS014855, in which the *hin* gene is truncated (PNCS009777 reference; *hin* appears absent at this location in the pangenome when *S.* Typhimurium LT2 is used as the reference); 2) complete deletion of *hin* with intact *fljAB* and adjacent DNA (isolate PNCS014870); 3) deletions of *hin* with point mutations in *fljB* (isolates PNCS015008 & PNCS014871); 4) loss of *hin* and *fljB*, while *fljA* is intact (isolate PNCS015158, BLAST hits to the *N*- and *C*-termini of *fljB* represent the regions of conservation of the *fljB* and *fliC* genes rather than the presence of *fljB* sequences); 5) isolates PNCS009991 (closed genome) and PNCS015370 (draft genome) have a deletion encompassing *hin* and *fljAB* through the accompanying upstream region containing phage-related genes; 6) isolate PNCS014863 (closed genome) has a deletion that includes truncation of STM2759 and deletion of five downstream genes comprising the *iroB-N* locus. Two isolates without the MREL, PNCS015054 (closed genome) and PNCS014854 (closed genome) were included for comparison (Fig 3). Both have deletions ending at an intact *hin* gene that are different than those associated with the MREL and are representative of isolates outside the global MREL-containing clone. All these deletions begin after or within ST2759. The FELS-2 prophage was missing in all these isolates, which may or may not have been associated with the event leading to lesions at *fljAB hin*. Absence of FELS-2 is seen in some *S.* Typhimurium isolates (unpublished observations) [47,48]. Insertion of the MREL is also associated with additional deletions (S1 Fig): 7) between STM2760 and *iroB*; 8) between STM2745 and *iroB* (PNCS014866); complex rearrangement ending at *iroB* (PNCS014867). These data fit a model in which IS26 or the MREL inserts relatively precisely in the genome (near *iroB*) and causes imprecise excision of adjacent DNA. The transmissible element may also cause rearrangements with or without excision of DNA, as shown by Boland and colleagues [17].

A Maximum-Likelihood (ML) phylogenetic SNP tree (S2 Fig) was constructed using SNVPhyl as described in the Materials and Methods and summarized in S1 SNVPhyl analysis. This tree contained 4 major and 8 minor clades, as well as additional branches containing one or very few isolates. This dendrogram is large and complex and does not lend itself well to quick, intuitive apprehension and comparison of data, though it serves as an excellent reference. The MST had congruent topology with the ML tree, exhibiting the same 4 major and 8 minor clades, as well as the smaller branches. Furthermore, there was a high concordance of isolate placement within the two trees. However, the ML tree was less discriminatory than the MST, so that there were multiple instances of groups of closely related isolates not being differentiated in the ML tree that showed good differentiation in the MST. MSTs were therefore used throughout. A MST was therefore constructed on the basis of wgMLST data in Bionumerics and used throughout the analysis. The UPGMA dendrogram is provided for reference (S3 Fig).

The MREL was predominantly found in a single large clade (Clade IV) within the population of *Salmonella* 4,[5],12:i:- characterized for this work, but it was also present in scattered isolates outside the large main clade of MREL-containing isolates (Fig 5, S3 Fig). These data suggest that there is continuing acquisition of the MREL in addition to expansion of the clade, consistent with data reported previously [30].

To address the possibility that the MREL may have been acquired repeatedly within Clade IV the presence or absence of *fljA*, *fljB*, *fljB* genes with point mutations, and *hin* genes has been identified within Clade IV of the UPGMA dendrogram (S4 Fig). MREL-positive isolates containing *hin* genes, especially, can be found in sub-clades that otherwise contain isolates lacking *hin* genes, and there are other configurations of gene presence, absence evident, and mutations

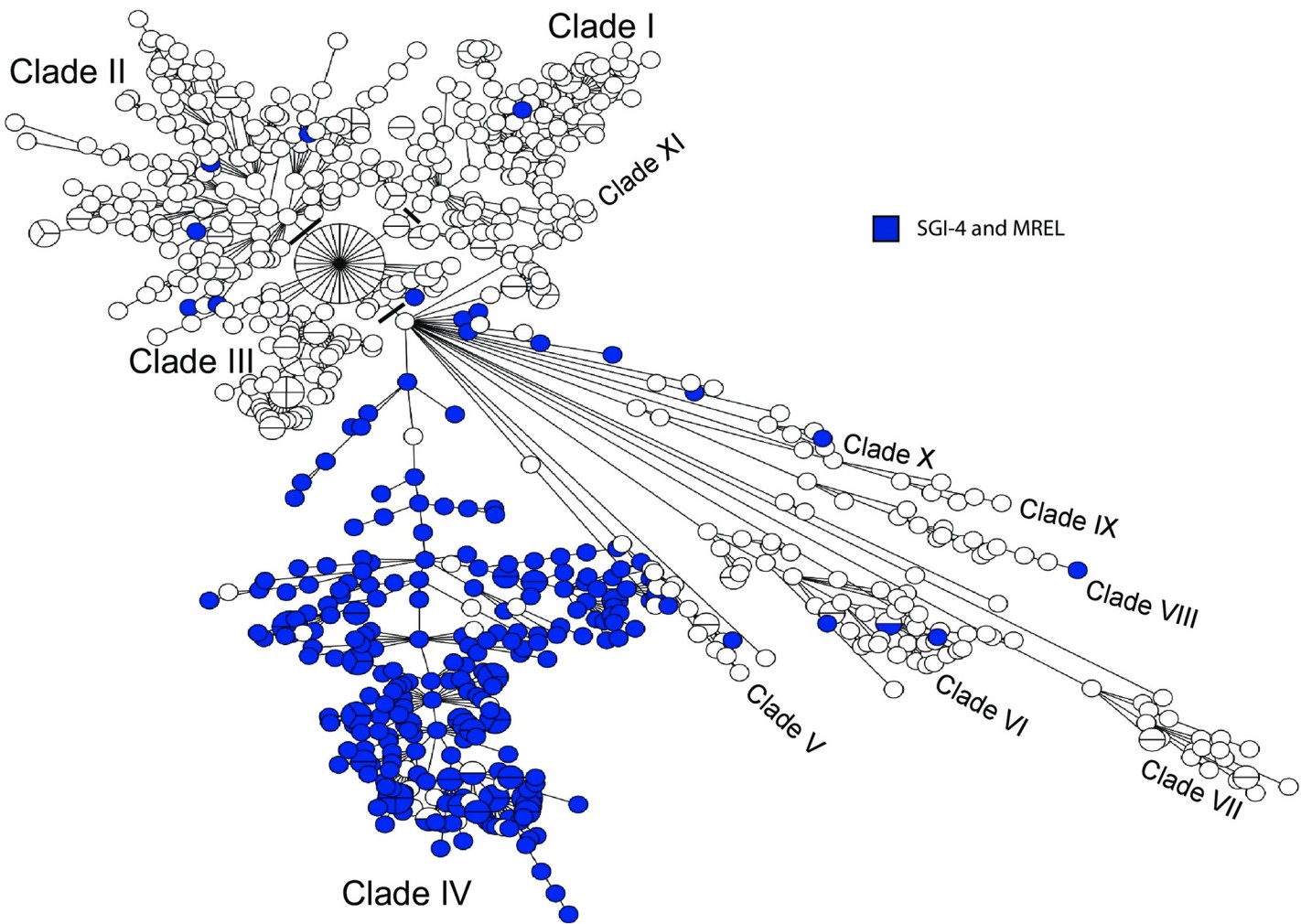

**Fig 5. Minimum spanning tree of wgMLST data showing the distribution of SGI-4 and the MREL.** The MST analysis was performed in Bionumerics v7.6.2. Four outliers were removed from the analysis to enable visualization of the tree structure.

present. The ML and UPGMA dendrograms presented in this work only allow inference of the relative relatedness of isolates, not their evolution. It is therefore possible that some of these isolates may have been clonally derived from isolates more distant in the dendrogram that also retain *hin*, with other changes arising through rearrangement of the chromosomal region. It is also possible that these isolates may have acquired their 4,[5],:12:i:- genotype through a separate acquisition of the MREL, which has the requisite elements to promote chromosomal integration. In our opinion the two possibilities cannot be distinguished with the extant data, and details of the evolution of *Salmonella* 4,[5],12:i:- populations containing the MREL and SGI-4 remains an open question.

## Plasmids carry genes contained in the MREL and SGI-4

Eight plasmids identified as having substantial identity with the chromosomal MREL in blastn searches of NCBI databases or in the scientific literature were compared with the sequences of three isolates with complete closed and finished genomic sequences in BLAST-based GView pangenome analysis using reference strain PNCS009777 as the reference (Fig 6). With the

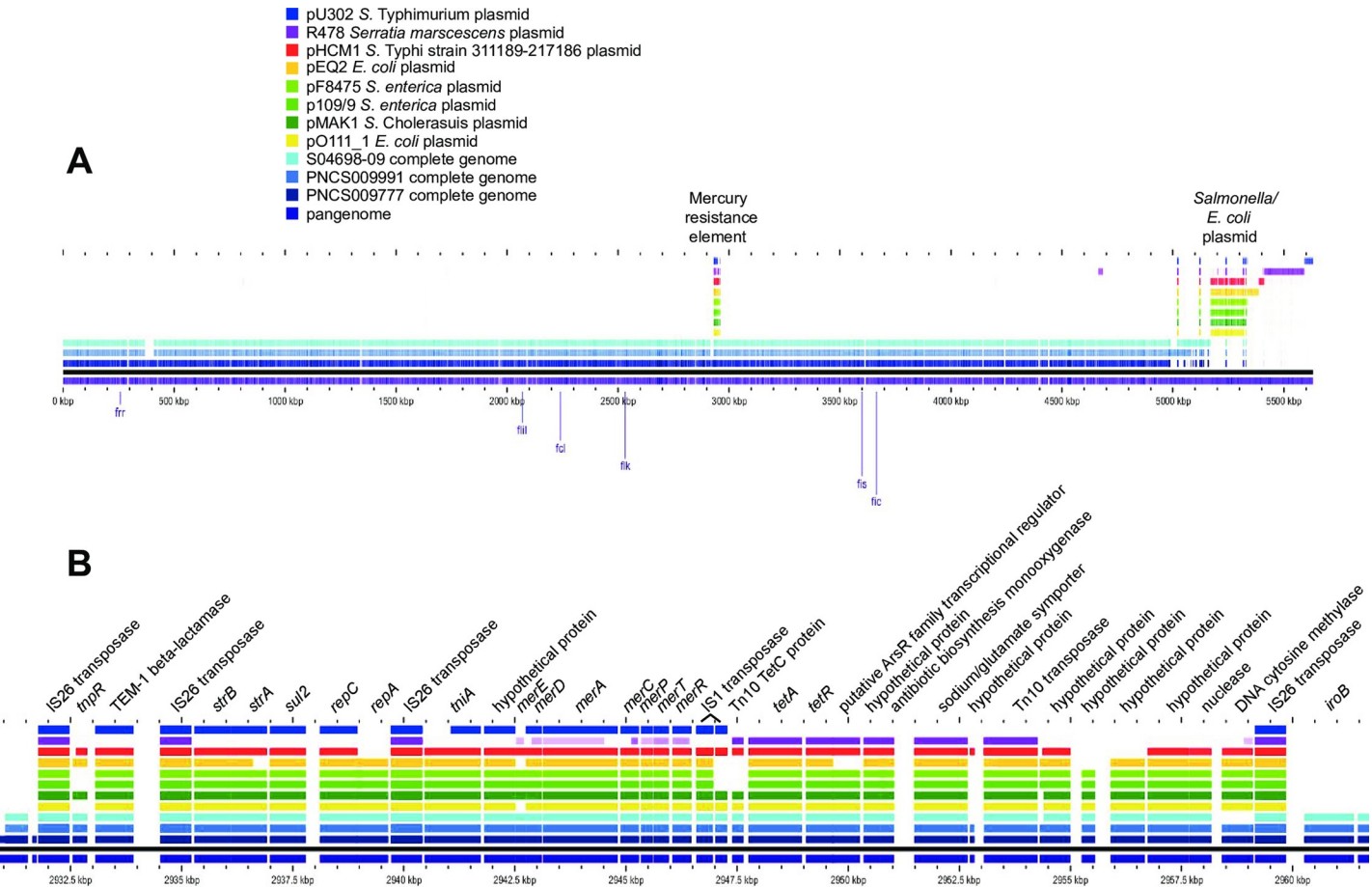

**Fig 6. Gene content of plasmids with MREL genes compared with chromosomal MREL from reference genomes.** Pangenome analysis was conducted using GView Server. A. Plasmids were compared with the entire genomes of reference isolates to show the location of MREL genes and identities of the plasmid backbones. B. The plasmid gene content was compared to the MRELs in reference genomes.

exception of plasmids R478 and pU302, which had very little identity of the plasmid backbone with other plasmids (Fig 6A, to the right of the closed genomes), the plasmid backbones had very similar gene content and high identity. These observations were consistent with the presence of three different MREL plasmids and the spread of at least one of the *E. coli/Salmonella* MREL-containing plasmids. There was loss of some MREL genes in plasmids from different isolates. The unique *Serratia marscescens* plasmid R478 (Accession no. BX664015.1) carries mercury resistance alleles with identities of less than 90% compared with those in the remainder of the plasmids and in the chromosomal MRELs (Fig 6B). It also carries the arsenate resistance genes *arsCBRH* [49,50] but these genes appear to have less than 90% identity to genes encoding arsenate resistance in SGI-. The MREL of plasmid pU302L (Accession no. NC_006816.1) from *S.* Typhimurium strain G8430 lacked all genes downstream of the mercury resistance gene region except for the final IS26 transposase (Fig 6B). Two plasmids, pO111_1 (Accession no. AP010961.1) from *E. coli* O111:H- strain 11128 and pMAK1 (Accession no. AB266440.1) from *S.* Cholerasuis strain L-2454 contained MREL gene content most closely related to chromosomal MRELs in isolates with complete genomes (Fig 6B). A large part of the MREL was previously shown to have 99% sequence identity with the *E. coli* plasmid pO111_1 [30]; however, pO111_1 lacked *tnpR* and *merE*. The gene encoding the Tn10 TetC

protein, one of two adjacent IS1 transposases, and *tnpR* were not present in *S. enterica* plasmids p109/9 (Accession no. KP899805.1) from strain 109/9 and pF8475 (Accession no. KP899804.1) from strain F9475 (Fig 6B). *E. coli* strain 63743 plasmid pEQ2 (Accession no. KF362122.2) has deletions of the *strA* and *merE* genes, as well as genes encoding hypothetical proteins and the Tn10 TetC protein. Finally, pHCM1 (Accession no. CP029645.1) from *S.* Typhi strain 311189–217186 has a truncated *tnpR* gene and lacks *repA* and two genes encoding hypothetical proteins. Overall, the deletions in these plasmid MRELs are limited within each plasmid but heterogeneous across the population.

Despite the high similarity of gene content in the MRELs of these plasmids there was considerable variability in the order of genes within the plasmid genomes (Fig 7) which was somewhat greater than the variability in MRELs from *Salmonella* 4,[5],12:i:- genomes (Fig 4). With the exception of variably present *merE*, the mercury resistance operon was present in all plasmid MRELs shown in Fig 7. Deletions of insertion elements were apparent, as were inversions and rearrangements of genes bounded by insertion elements. Plasmids pF8475 and p109/9 had insertions of additional antimicrobial resistance genes (Fig 7, Greek lettering). These

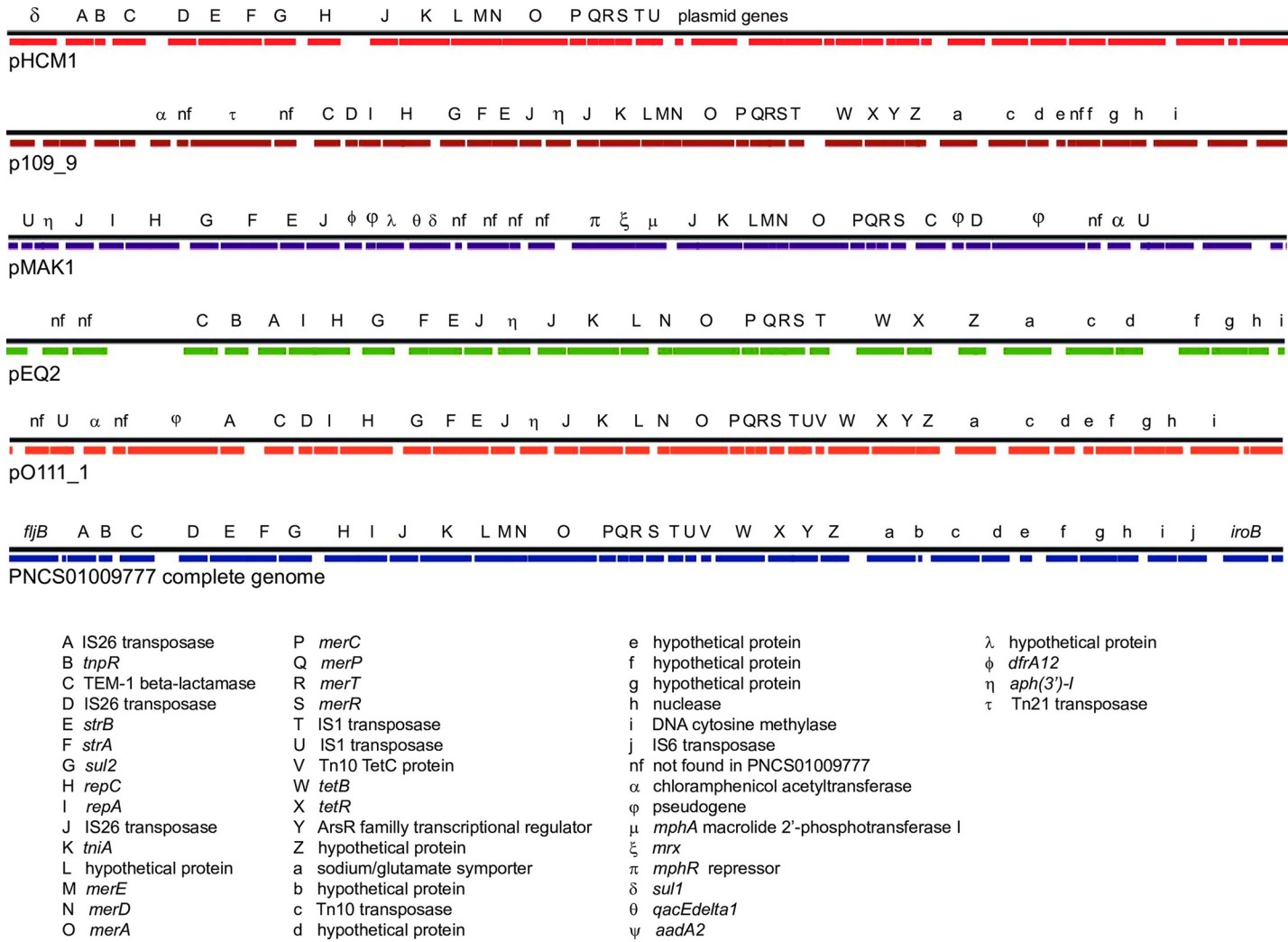

**Fig 7. Synteny and variability of the MREL in previously characterized plasmids compared with the MREL of reference isolate PNCS009777.** This figure was constructed the same way as Fig 4.

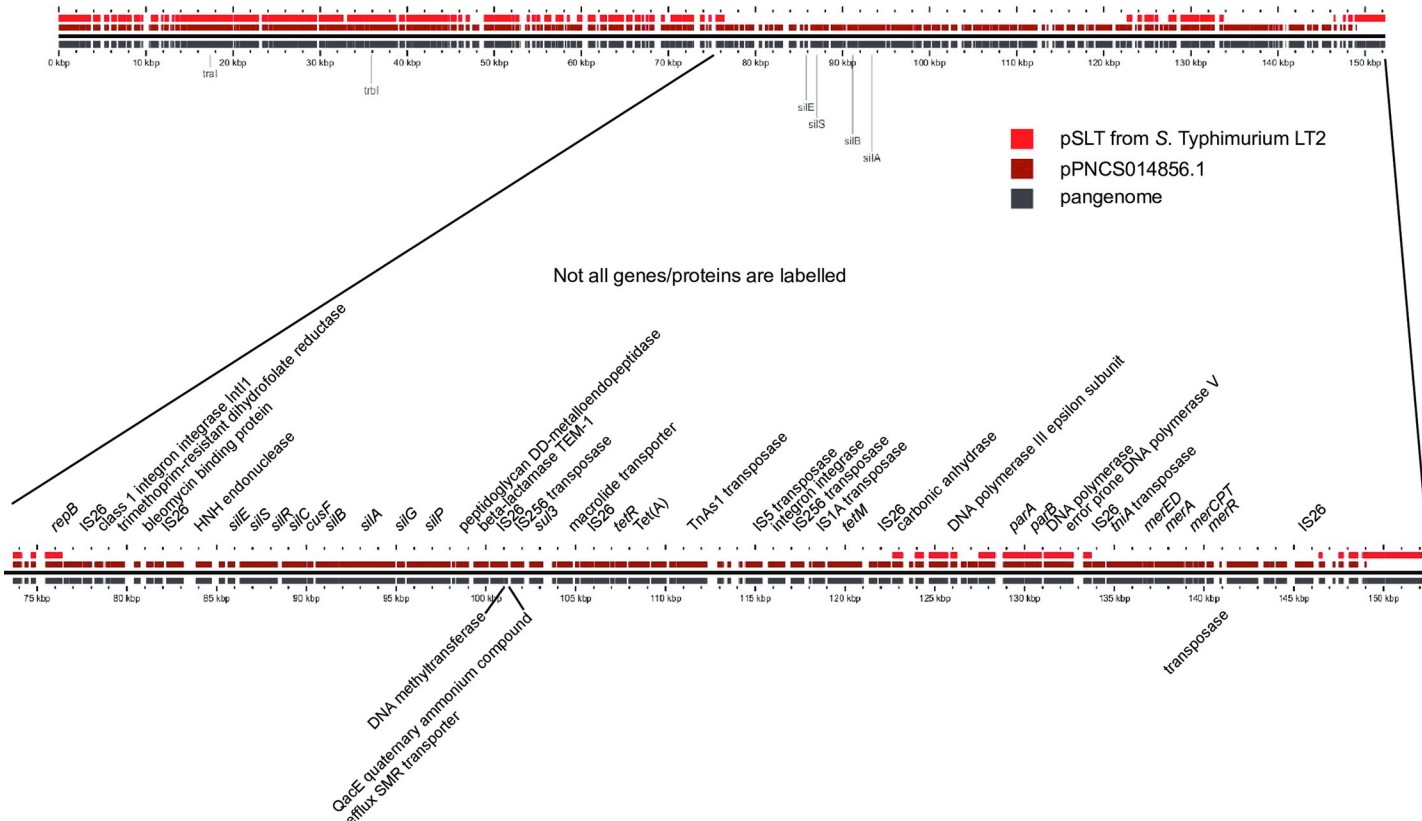

**Fig 8. GView pangenome comparison of the genomes of plasmid pSLT and pPNCS014856.1.** Almost all genes of pSLT are contained within pPNCS014856.1, which has indels associated with genes from SGI-4 and the MREL, as well as genes from other sources. pPNCS014856.1 was used as the reference to show that the mercury resistance genes from the MREL are inserted at a different location within pSLT than genes conferring antibiotic resistance from the MREL and silver resistance from SGI-4.

observations suggest evolution within a single plasmid population, and are consistent with the known variability of antibiotic resistance genes associated with the MREL [51].

Plasmid pPNCS014856 was identified in this work as a plasmid carrying genes associated with the MREL and the Copper Homeostasis and Silver Resistance Island (CHASRI) [52] integrated into a pSLT plasmid (Fig 8). Complete or partial variants of this plasmid were not detected in the isolates carrying the MREL and SGI-4 (compare Figs 9 & 5) nor in any isolates carrying the pSLT plasmid (compare Figs 9 & 10). The mechanism by which pSLT is excluded from isolates with the MREL and SGI-4 is still not clear.

IS26 may mediate introduction of MREL genes or regions into additional plasmids present within an isolate to create a range of plasmids containing genes from the element. Plasmids with genes from the MREL (above) do not have the IncFIB(S) and IncFII(S) incompatibility types characteristic of the pSLT plasmids, and instead have: 1) IncFIA, IncHIA, and IncHI18 (R27) in plasmids pEQ2, p109/9, pF8475, pHCM1, & pO111_1; 2) IncHIA, and IncHI18(R27) in plasmid pMAK1; 3) IncFIA & IncFIB(AP00918) in plasmid pU302; 4) IncH12 & IncH12A in plasmid R478. The aggregate data are consistent with a population in which the chromosomal MREL and plasmids are capable of exchanging genetic material. Furthermore, IncA/C plasmids from poultry isolates from six different *Salmonella* serotypes carried the mercury resistance operon plus several of the antimicrobial resistance determinants present in chromosomal MRELs and plasmids [53].

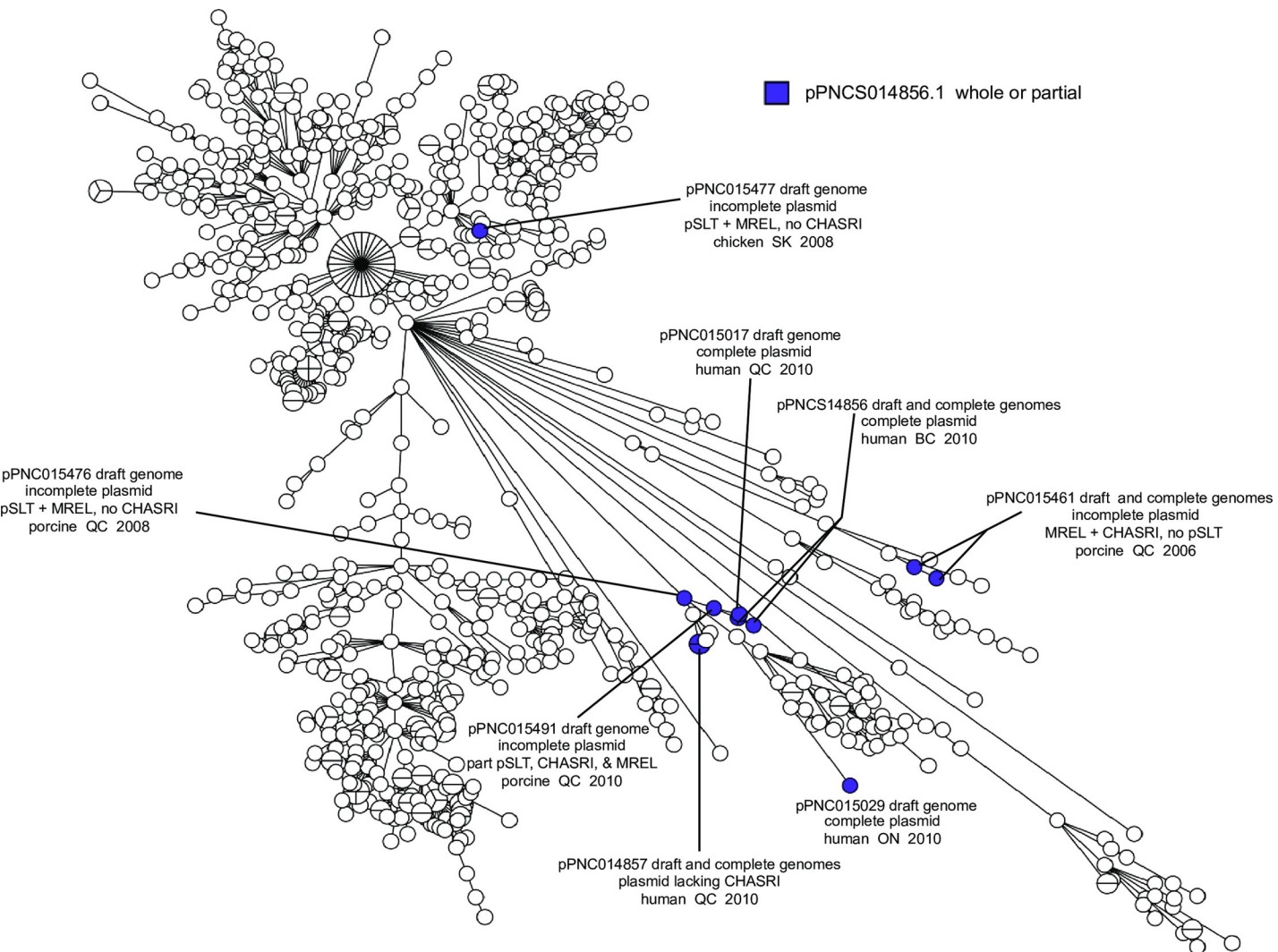

**Fig 9. Minimum spanning tree of wgMLST data showing the distribution of pPNCS014856.1 in the population of isolates used in this study.** Plasmids associated with complete sequences were identified as a consequence of genome completion and used to probe the entire *Salmonella* 4,[5],12:i:- population under study. The MST analysis was performed as for Fig 5.

## SGI-4 has limited heterogeneity

SGI-4 (Fig 11) was most frequently inserted between genes encoding AraC family transcriptional regulator/GadX (STM4320) and tRNA-Phe (STM4321; see Figs 1 and 12). In isolates PNCS014852 and PNCS014866 SGI-4 was located between tRNA-Phe (STM3116) and STM3117. It was also found in additional *Salmonella* genomes from GenBank (Accession numbers: SO4698-09, NZ_LN999997; 81741, CP019442.1; DA34821, CP029567.1; DA34837, CP029568.1; TW-Stm6, CP019649), which were the only ones exhibiting a max score of 1.551e+05, 100% query coverage, and 99–100% identity in a search done Jan16, 2019 using the region from our isolate PNCS009777 as the query sequence.

SGI-4 has previously been characterized as an Integrated Conjugative Element (ICE) capable of excision, circularization, and transmission to naïve strains with enhanced frequency in response to mitomycin C and anaerobiosis, with integration at the *pheV*-tRNA or *pheR*-tRNA loci at the same genomic location in the new strain [28,54]. Several of the other genomes and

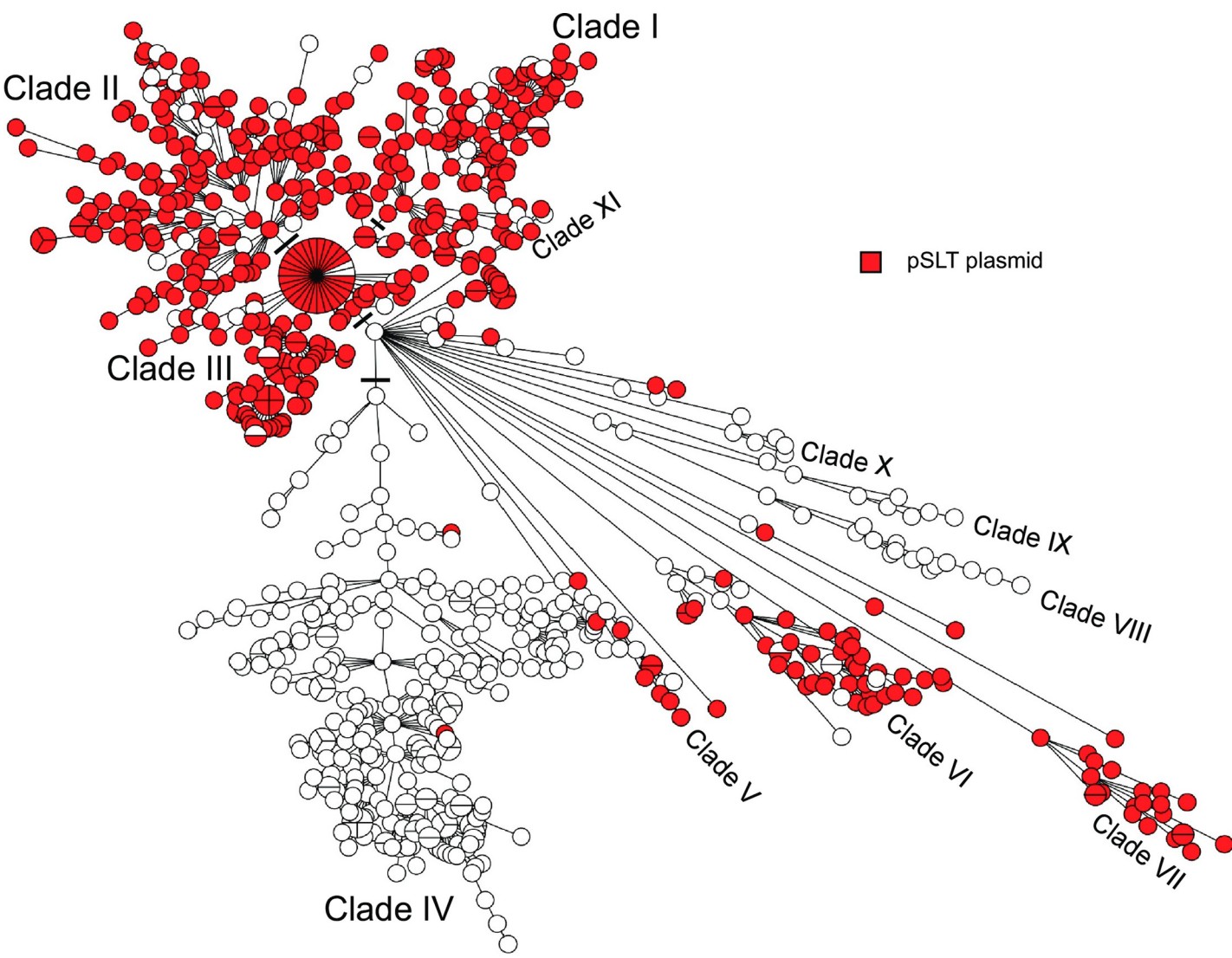

**Fig 10. Minimum spanning tree of wgMLST data showing the distribution of the pSLT plasmid in the population of isolates used in this study.** The MST analysis was performed as for Fig 5.

all 20 plasmids returned from this search contained only the CHASRI, leading us to speculate it may be subject to more intense selective pressure than the remainder of SGI-4. The origin of SGI-4 therefore does not appear to be associated with any plasmid characterized to date. This is consistent with previous work [16]. Structurally, SGI-4 shows less heterogeneity than the MREL (Fig 12), with isolates containing either the entire island or only the CHASRI. Previously characterized isolate TW-Stm6 (NCBI accession no. NZ_CP019649.1) carries both the MREL (Fig 13A) and SGI-4 (Fig 13B) in its genome. Plasmid pSTM275 from this strain carries some antimicrobial resistance genes from the MREL and the CHASRI from SGI-4 (Fig 13). However the plasmid genes have less than 99% identity with the SGI-4 CHASRI genes: *silE*, 91% identity; *silS*, 97%; *silR*, 98%; *silC*, 96.5%; *silF*, 96%; *silB*, 97%; *silA*, 89%; *pcoG*, 97%; *pcoE1*, 97%; *pcoB*, 94.5%, *pcoC*, 97%; *pcoD*, 97%; *pcoS*, 97%; *pcoE2*, 90.5%. It thus appears that this plasmid CHASRI was acquired from a source other than the chromosomal SGI-4 in strain TW-Stm6. The CHASRI has been identified previously as a 32.4 kb Tn7 family transposon

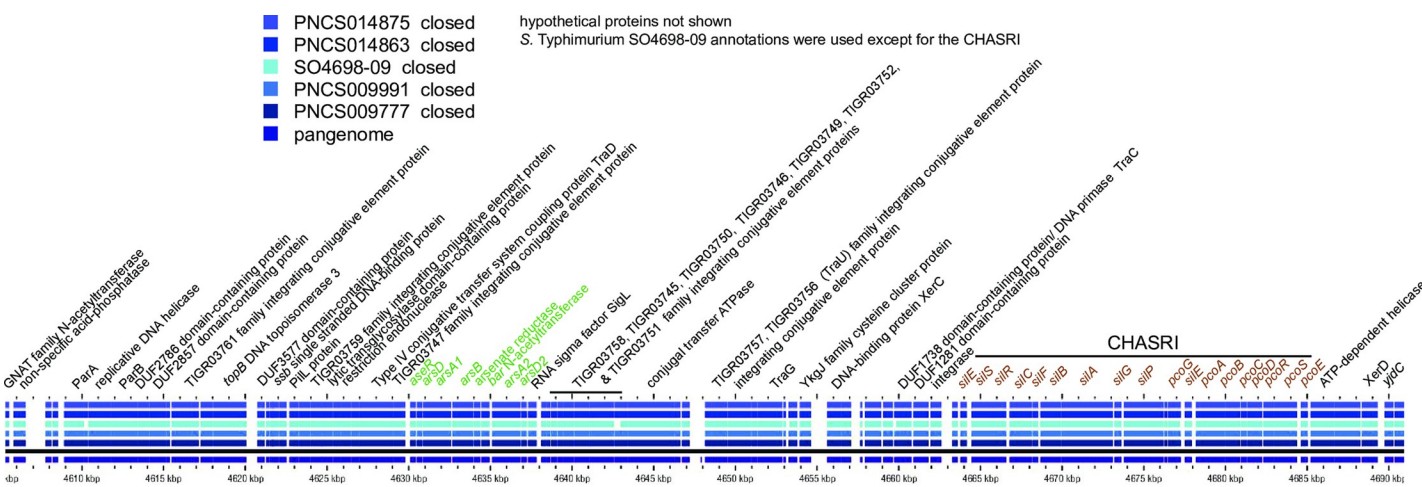

**Fig 11. Structure of SGI-4 in four isolates for which closed genome sequences were obtained.** The comparison was done using GView Server with the "Pangenome analysis" settings and the figure was annotated in Adobe Illustrator CS6.

**Fig 12. Heterogeneity of SGI-4 in selected, representative isolates.** The figure was prepared as for Fig 11.

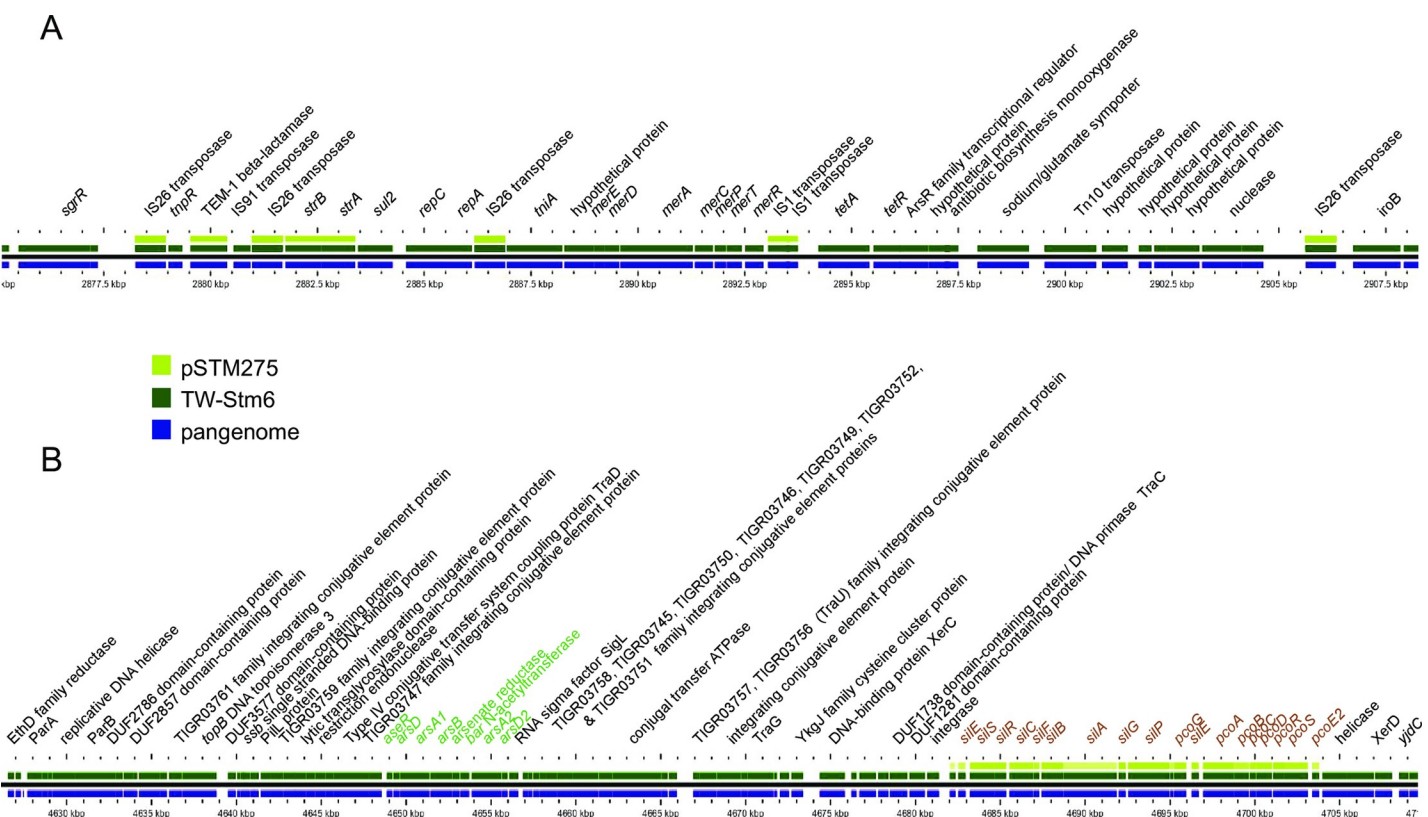

**Fig 13. Comparison of the SGI-4 and MREL content of plasmids pSTM275 and TW-Stm6.** GView pangenome analysis using TW-Stm as the reference genome. Data were analyzed as in Figs 5–7.

carrying silver and copper resistance loci [50,52,55]. The presence of integrase and recombinase genes flanking the CHASRI suggests a possible mechanism by which this region could be mobilized independently of the remaining SGI-4. Despite carrying plasmid mobilization genes, SGI-4 did not have a locus conferring incompatibility. This may be an adaptation to increase the stability of the island, which may have been assembled through horizontal transfer of mobile elements [22].

Isolates with the CHASRI only were not found in the large clades with the majority of the isolates containing the complete SGI-4, but were instead distributed in different parts of the dendrogram. We think that SGI-4 may have been assembled in a *Salmonella* chromosome and may be mobilizable via conjugation into new chromosomes utilizing the genes present on the island (Fig 7) consistent with the suggestion by Petrovska and colleagues [16] that SGI-4 may have originated by integration of a plasmid, as well as with data indicating that SGI-4 is self-transmissible [28,52]. Alternately, genes from the SGI-4 and MREL may be found extrachromosomally in plasmids (see our discussion above; [33]). Within the Spanish clone *sil* and *merA* were detected in 110–220 kb non-transferable IncA/C plasmids, while in the Southern European clone *sil* was in 110–140 kb non-transferable IncR plasmids [33]. However, it is not clear whether *sil* represented SGI-4 or just the copper/silver resistance transposon in a plasmid or chromosomal location. It is also not clear how the *merA* and *sil* genes were acquired by the plasmids under study. Care should therefore be taken when interpreting the results of PCR or other single gene assays showing the presence of heavy metal resistance genes, as well as the presence of these genes within draft genomes.

A comparison of isolates containing complete SGI-4 or the copper/silver element shows a very close, though not absolute, correspondence with isolates containing the either the full MREL or only the mercury resistance operon. The close correspondence of the MREL and SGI-4 may have come about due to strong selection. The enhanced copper resistance afforded by SGI-4 [27], the heavy metal resistance of the MREL, and the additional antimicrobial resistance of both elements likely mediate the necessary strong selection.

## Partially overlapping Canadian and global clades contain the MREL and SGI-4

Of the 811 *Salmonella* 4,[5],12:i:- genomes analyzed in this study, 247 (30%) carried an intact or partial MREL and/or SGI-4 in the chromosome. Isolates with at least one of the chromosomal elements for the most part constituted a well-defined clade, as discussed earlier (Fig 5) that was present mostly in isolates from Ontario and Québec but with a subclade containing mostly isolates from western Canada (compare Figs 5 and 14). Both elements were found in

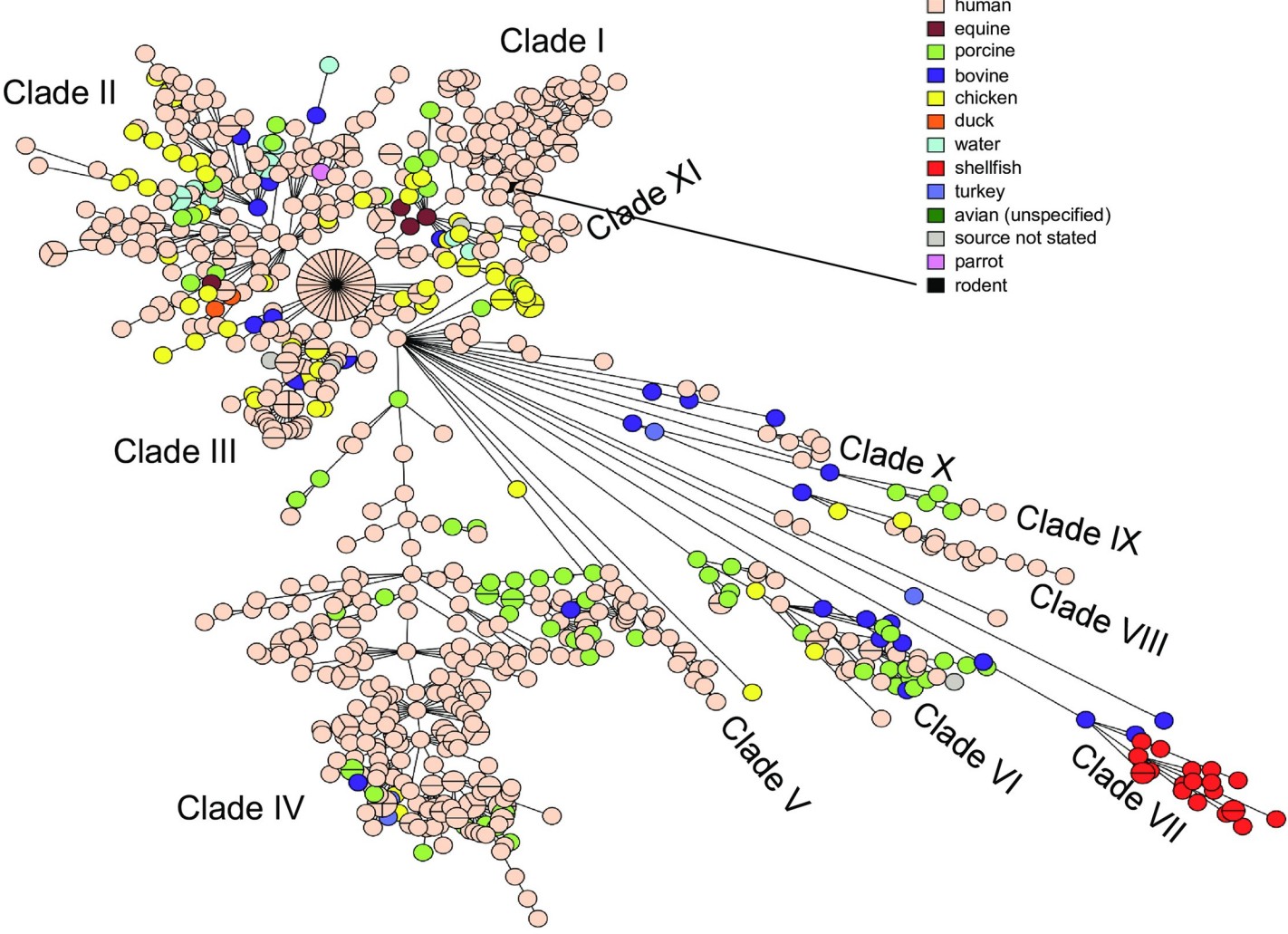

**Fig 14. Minimum spanning tree of wgMLST data showing the distribution of human and non-human sources associated with Canadian isolates.** The MST was prepared as for Fig 5.

214/247 (87%) of these isolates. This was consistent with previous results indicating that both the MREL and SGI-3 elements were frequently detected together in Spanish and European *Salmonella* 4,[5],12:i:- clones and that they could be structurally heterogeneous [33]. These authors also determined that the MREL and SGI-4 were associated with different plasmids, as noted earlier.

The presence of SGI-4 or the MREL has been characterized as defining a well-defined clade within the *Salmonella* 4,[5],12:i:- population structure in both the UK (Public Health England, or PHE) [16] and the US [15]. A MST tree was produced using our Canadian data and sequence data retrieved from the NCBI Sequence Read Archive using metadata in published papers [15,16]. The multi-clade visualization obtained (Fig 15; S5 Fig) indicates that Canadian, US, and English/European isolates with SGI-4 and the MREL are distributed in different proportions in different clades. Because of the density of data represented in this diagram and the difficulty of assigning membership to clades inherent in the reduction of a three-dimensional representation, we also looked at the clade structure in the original dendrogram (S3 Fig) used to construct the MSTs in BioNumerics. In this dendrogram there several major clades labelled A—H. Clades A and B contain isolates from all regions, while clade C has isolates only form Canada and England/Europe. With very few exceptions, isolates from the US and Canada comprise clades D–G, while Clade H is composed of isolates from Canada, predominantly Québec. Isolates within clade E are mostly ST2379 with a small ST 34 sub-clade, whereas isolates in other clades are almost all ST34. ST2379 differs from ST34 only at the *hemD* allele. These data suggest clonal origin and expansion of clade D isolates in North America, consistent with previous observations made by Elnekave and colleagues [15]. Isolates sequenced by Public Health England were obtained earlier in time than those from either the US or Canada, and from a greater diversity of sources in both the UK and the US than Canada. It is not clear whether these differences represent a differential spread into animal populations within in different jurisdictions, different surveillance systems, or both.

## The MREL and SGI-4 of Canadian non-human isolates are strongly associated with porcine sources

Non-human *Salmonella* 4,[5],12:i:- were acquired by the Canadian FoodNet (previously C-EnterNet) program as part of active surveillance for human and non-human foodborne bacterial pathogens in sentinel sites [56,57] and 192 were sequenced as part of this research. Additional isolates from non-human sources were obtained elsewhere. About 22% of these Canadian isolates from non-human sources carried the MREL and/or SGI-4 and were associated with porcine sources (see Table 1). Most of these isolates were recovered in Ontario and Québec, perhaps because half of the total non-human *Salmonella* 4,[5],12:i:- were from these two provinces or because of idiosyncrasies in the surveillance system for non-human isolates. Previous work has indicated that there is a very strong correlation of heavy metal resistant *Salmonella* 4,[5],12:i:- with porcine sources that has been attributed to the use of heavy metals, especially copper, as growth promoters in the hog industry [29,33,58,59]. The disease lesions resulting from *Salmonella* 4,[5],12:i:- infection of pigs is similar to that produced by *S*. Typhimurium [5]. Danish strains with the resistance phenotype encoded by the MREL predominate in pork but are also recovered less frequently from cattle and occasionally from poultry [60]. MREL-containing *Salmonella* 4,[5],12:i:- from humans, pigs, and pork had identical traits [51], strongly implicating pigs and pork as a source of human infection and outbreaks [61]. SGI-4 was present in human isolates from within and outside the ST34 clone, though none of the chicken and turkey isolates within the ST34 clone were SGI-4 positive (S3 Fig). This

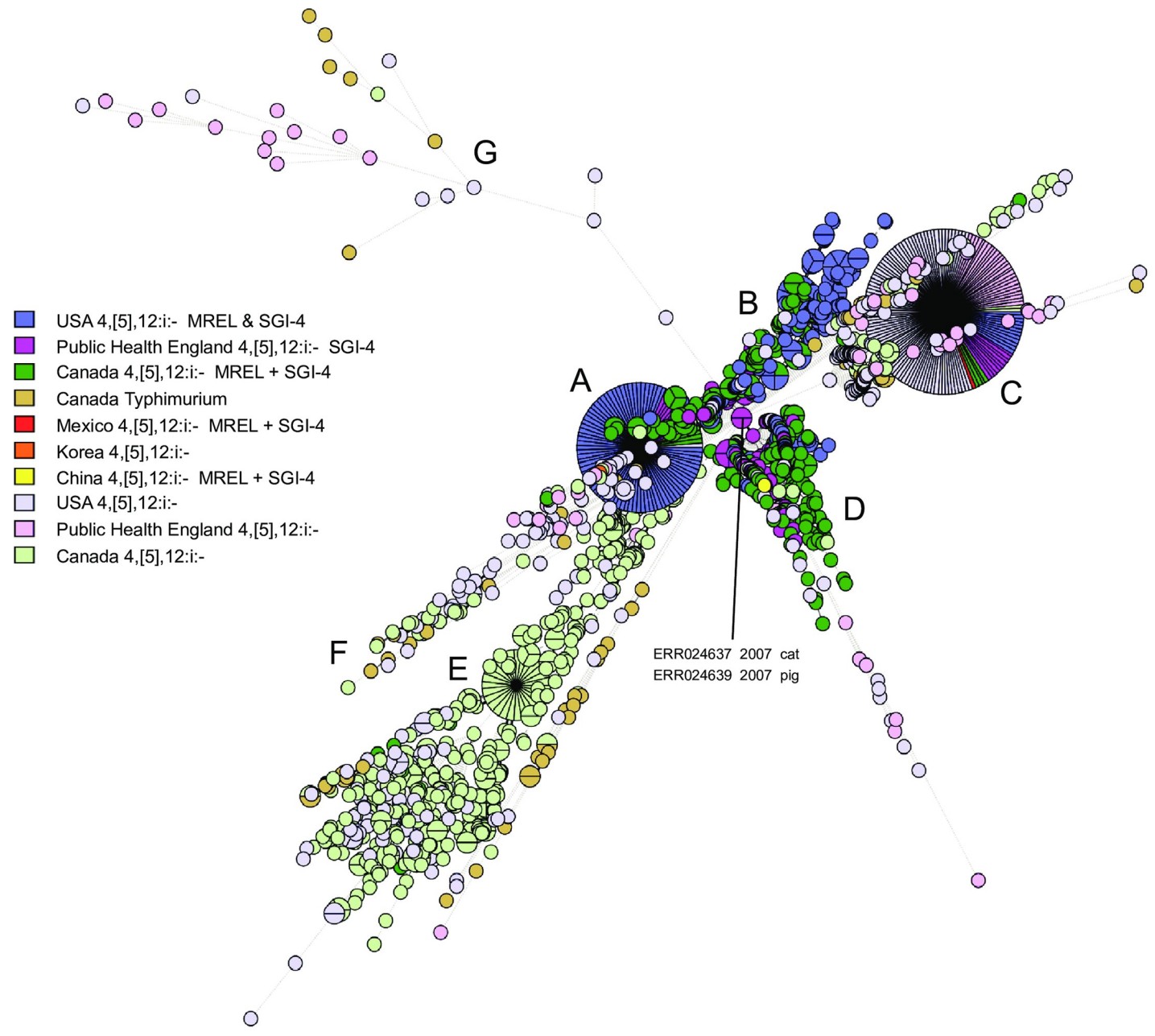

**Fig 15. Minimum spanning tree of wgMLST data showing the distribution of SGI-4/MREL-containing isolates within the global *Salmonella* 4,[5],12:i:-population analyzed.** The MST was prepared as for Fig 5. The presence of SGI-4 and/or the MREL for the US and Public Health England isolates was inferred from the metadata in the respective published papers [14,15]. For the US data the presence of the MREL was indicated by the detection of potential RR1, RR2, and RR3 sequences (S1 Table in [14]). The presence of SGI-4 was not indicated. The PHE data clearly indicated the presence of SGI-4 (labelled SGI-3, see Technical Appendix 2, Fig 3 in [15]).

suggests that the chicken and turkey isolates either lost SGI-4 due to lack of selection or were not part of the global clone containing this chromosomal element.

Continued surveillance of both human and non-human sources of *Salmonella* 4,[5],12:i:-isolates is valuable for identification of animal or food sources of the organism. Knowing the establishment(s) from which the isolates were obtained, as was the case here, provides critical

**Table 1. Non-human isolates carrying the MREL and/or SGI-4 versus total number of isolates tested, by source and province.**

| Province of origin | Porcine | Bovine | Equine | Chicken | Turkey | Other avian | Shellfish | Water | Total |
|---|---|---|---|---|---|---|---|---|---|
| British Columbia | 1/2 | 0/4 | - | 0/14 | - | 1/1 | - | | **2/21 (10%)** |
| Alberta | 0/4 | 0/3 | - | 1/14 | - | - | - | | **1/21 (5%)** |
| Saskatchewan | 0/3 | 0/1 | 0/3 | 0/6 | - | 0/2 | - | | **0/15** |
| Manitoba | 2/3 | 0/4 | - | 0/6 | - | - | - | | **2/13 (15%)** |
| Ontario | 12/14 | 1/13 | 0/1 | 1/19 | 2/4 | 0/2 | - | | **16/53 (31%)** |
| Québec | 17/40 | 1/2 | - | 0/4 | - | 0/1 | - | | **18/47 (38%)** |
| New Brunswick | - | - | - | - | - | - | 0/14 | | **0/14** |
| Nova Scotia | - | - | - | - | - | - | 0/2 | | **0/2** |
| PEI | 0/2 | - | - | - | - | 0/1 | 0/4 | | **0/7** |
| Not known | - | | | | | | | 0/16 | **0/16** |
| **Total** | **32/68 (47%)** | **2/27 (7%)** | **0/4** | **2/63 (3%)** | **2/4 (50%)** | **1/7 (14%)** | **0/20** | **0/16** | **45/209 (26%)** |

information that complements human surveillance data. Prospective surveillance of food and animals is valuable for identification of the ways organisms are transmitted to the human population, as well as for outbreak investigation and control. Further investigations are required to determine whether interventions can be developed to reduce the contribution of heavy metal-resistant *Salmonella* to human morbidity by restricting use of heavy metals as growth promoters.

## Conclusions

*Salmonella* 4,[5],12:i:- isolates containing SGI-4 and the MREL are only part of the total Canadian 4,[5],12:i:- population characterized in this work. These elements were already present in isolates in 2008 and were found until the study end in 2016, though the non-random isolate selection makes it difficult to determine whether they were increasing within the *Salmonella* 4,[5],12:i:- population. They are found in a clade of human isolates and were strongly associated with porcine sources and closely related to the European and American clonal expansion of *Salmonella* 4,[5],12:i:- resulting from the acquisition of SGI-4. However, there appears to be some expansion of clones unique to Canada. Sequence data associated with the *iroB-hin-fljAB* loci indicates that there have been multiple introductions of the MREL that frequently result in different genetic scars that will be useful in strain characterization for investigation public health events like case clusters and outbreaks. Our data support previous observations that both SGI-4 and the MREL are mobile elements. Further work will assess the proportion of *Salmonella* 4,[5],12:i:- isolates containing SGI-4 and MREL during the last few years, when all Canadian *Salmonella* genomes were sequenced by PulseNet Canada.

## Supporting information

**S1 Fig. Variability of the MREL in eleven closed genomes.** STM numbers referring to locus designations in *S.* Typhimurium LT2 aid in identification of the genomic location of MREL insertion. Regions associated with inversion or rearrangement are shown in red. The comparison was done using GView Server with the "pangenome analysis" settings and the figure was annotated in Adobe Illustrator CS6.
(PDF)

**S2 Fig. Maximum likelihood SNP tree.** The tree was constructed using SNVPhyl and annotated in FigTree v1.4.4. It was rooted with *S.* Thompson strain PNCS015102 and also contained *S.* Heidelberg strain PNCS015430 to validate the structure. The order of annotation at

each node tip is: isolate designation, province of origin, source, and genetic lesion causing the monophasic phenotype. The scale bar is at the bottom of the dendrogram.
(PDF)

**S3 Fig. wgMLST UPGMA dendrogram of all Canadian *Salmonella* 4,[5],12:i:- isolates.** Isolates containing SGI-4 and the MREL are highlighted. Selected metadata are included. The wgMLST dendrogram was generated in BioNumerics 7.6.2.
(PDF)

**S4 Fig. wgMLST UPGMA dendrogram with Clade IV Canadian *Salmonella* 4,[5],12:i:- isolates.** This dendrogram correlates SGI-4 and the MREL presence with *fljAB hin* gene presence, absence, or mutation, as well as with location in the dendrogram. The wgMLST dendrogram was generated in BioNumerics 7.6.3.
(PDF)

**S5 Fig. wgMLST UPGMA dendrogram showing major clades of from Canada, the US, and England/Europe.** SGI-4 and MREL-containing isolates are differentiated from those without these elements. The Canadian data are from this study, while the US data are from genomes referenced in Elnekave and colleagues [14] and the Public Health England data are from genomes referenced in Petrovska and colleagues [15]. The dendrogram was produced using wgMLST data from assembled genomes using BioNumerics 7.6.3.
(PDF)

**S1 File. Data describing the SNP analysis and maximum likelihood tree construction were obtained from the SNVPphyl pipeline.**
(DOCX)

**S1 Spreadsheet. Selected isolate metadata.**
(XLSX)

**S2 Spreadsheet. Quality statistics for WGS data for assembled genomes.** The data were generated using tools within BioNumerics version 7.6.2.
(XLSX)

## Acknowledgments

We acknowledge the PulseNet Canada (PNC) Steering Committee and provincial Public Health laboratories (PHLs) from all Canadian provinces and territories for characterization of isolates, collection and curation of metadata, sending isolates to the NML Winnipeg, and granting permission to use isolates and associated metadata. Thanks to Lorelee Tschetter for managing all interaction and inquiries associated with the PNC Steering Committee and provincial PHLs. Genome sequencing was done within and by DNA the sequencing facility in the Genomics Core Facility at NML Winnipeg. We would like to thank Morag Graham, Director of the facility, Brynn Kaplen, Christine Bonner, Geoff Peters, Kim Melnychuk, Erika Landry, Shari Tyson, and Vanessa Laminman, for preparation of libraries and all other steps in sequencing, We would also like to specifically thank Shaun Tyler in the Genomics Core Facility for his input and labour during the pilot study comparing MiSeq and NextSeq sequencing. Thanks to Guangzhi Zhang for DNA template preparation and to Chantal Munyuza for help with elucidating the distribution of plasmids within the Canadian isolate draft genomes. Many thanks also to Chris Yachison for sharing relevant results and discussions resulting from his work on SISTR. Thanks to Sandeep Tamber and Catherine Carrillo for a critical reading of the manuscript. We acknowledge the work of the Bioinformatics Core of NML Winnipeg, led by

Dr. Gary van Domselaar, for developing bioinformatics tools that make it possible for biologists to analyze whole genome sequence data, and especially Aaron Petkau for the implementation of GView Server and the SNVPhyl pipeline.

## Author Contributions

**Conceptualization:** Clifford G. Clark.

**Data curation:** Clifford G. Clark, Chrystal Landgraff, Frank Pollari, Stephen Parker, Victor P. J. Gannon, Roger Johnson.

**Formal analysis:** Clifford G. Clark, Chrystal Landgraff.

**Funding acquisition:** Clifford G. Clark, Roger Johnson.

**Investigation:** Clifford G. Clark, Chrystal Landgraff, James Robertson.

**Methodology:** Clifford G. Clark, Chrystal Landgraff.

**Project administration:** Clifford G. Clark, Roger Johnson.

**Resources:** Frank Pollari, Stephen Parker, Victor P. J. Gannon, John Nash.

**Supervision:** Clifford G. Clark, Celine Nadon, Roger Johnson, John Nash.

**Validation:** Clifford G. Clark, Chrystal Landgraff, James Robertson.

**Visualization:** Clifford G. Clark.

**Writing – original draft:** Clifford G. Clark.

**Writing – review & editing:** Clifford G. Clark, Chrystal Landgraff, James Robertson.

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
