## [Decision Letter · Decision Letter 0]

15 Apr 2020

PONE-D-20-06506

Distribution of heavy metal resistance elements in Canadian Salmonella 4,[5],12:i:- populations and association with the monophasic genotypes and phenotype

PLOS ONE

Dear Dr. Clark,

Thank you for submitting your manuscript to PLOS ONE.  My apologies for the delay in the review process. A secured a single reviewer who was very positive about the manuscript, but had a few major/minor concerns about data presentation and interpretation. 

Therefore, we invite you to submit a revised version of the manuscript that addresses the points raised during the review process.

We would appreciate receiving your revised manuscript by May 30 2020 11:59PM. To enhance the reproducibility of your results, we recommend that if applicable you deposit your laboratory protocols in protocols.io, where a protocol can be assigned its own identifier (DOI) such that it can be cited independently in the future. For instructions see: http://journals.plos.org/plosone/s/submission-guidelines#loc-laboratory-protocols

We look forward to receiving your revised manuscript.

Kind regards,

Nicholas J Mantis

Academic Editor

PLOS ONE

Reviewers' comments:

Reviewer's Responses to Questions

**Comments to the Author**

1. Is the manuscript technically sound, and do the data support the conclusions?

Reviewer #1: Yes

2. Has the statistical analysis been performed appropriately and rigorously? 

Reviewer #1: N/A

3. Have the authors made all data underlying the findings in their manuscript fully available?

Reviewer #1: Yes

4. Is the manuscript presented in an intelligible fashion and written in standard English?

Reviewer #1: Yes

5. Review Comments to the Author

Reviewer #1: This manuscript is concerned with the monophasic variant of Salmonella serovar Typhimurium and the associated acquisition of genes conferring metal resistance. This association has been established previously, but the manuscript examines it in detail in Canadian isolates. The analysis is straightforward, but I do have some comments on it and some points about presentation.

1. The "phylogenetic" trees in the main text are minimum spanning trees. I have seen this before, and apparently the bionumerics software does it, but this is not a good choice for reconstructing phylogeny. The method forces every internal node to correspond to an isolate, which does not correspond to reality. The Supporting Information makes use of UPGMA. Because it produces ultrametric trees, this is generally not a good method. It is especially inappropriate for closely-related strains isolated at different times. Neighbor joining, minimum evolution, and least squares are a few methods that would be reasonable.

2. It is not clear to me whether the authors are suggesting multiple independent acquisitions within Clade IV. If so, why is this hypothesis favored over a single acquisition followed by different rearrangements?

3. I was initially confused by the use of "dcm". This is the name of a gene present in most Salmonella, which is different from the gene referred to here.

4. Line 78: "the remaining iroB genes" There is only one iroB gene, right? Should this be "iro genes"? Or perhaps "iroB operon genes"?

5. Line 92: "evade selection" is odd here. In fact it is selection that is presumably maintaining these elements. I imagine that the authors mean that selection against the element due to the cost of maintaining it is overcome by selection for resistance.

6. Line 125. If these are on a plasmid in the Spanish clone, does that mean that they are not involved in the monophasic phenotype? If so, this distinction would be worth making.

7. Line 138 "indels encoding heavy metal resistance". Strange way to put it. An insertion can encode resistance, but a deletion cannot.

8. Line 149 "Isolate numbers were replaced with randomly generated numbers..." Why? Does Supporting spreadsheet 1 associate the published numbers with biosample IDs, or only the original numbers?

9. Line 243 Extra "within the"

10. Line 327 I would not say that this information is "in addition to" phylogenetic analysis. Whether or not it is formally used that way, it is phylogenetic information that is distinct from the MLST genotypes.

11. Fig. 7 "Complete genome". A more relevant, informative label would be "Chromosome".

12. I might state briefly what fljA, fljB, and hin are in the Introduction.

6. PLOS authors have the option to publish the peer review history of their article (what does this mean?). If published, this will include your full peer review and any attached files.

Reviewer #1: No

---

## [Author Response · Author response to Decision Letter 0]

18 Jun 2020

Response to editor comment:

All instances of “data not shown” have been removed. All data are available.

Review Comments to the Author

Reviewer #1: This manuscript is concerned with the monophasic variant of Salmonella serovar Typhimurium and the associated acquisition of genes conferring metal resistance. This association has been established previously, but the manuscript examines it in detail in Canadian isolates. The analysis is straightforward, but I do have some comments on it and some points about presentation.

1. The "phylogenetic" trees in the main text are minimum spanning trees. I have seen this before, and apparently the bionumerics software does it, but this is not a good choice for reconstructing phylogeny. The method forces every internal node to correspond to an isolate, which does not correspond to reality. The Supporting Information makes use of UPGMA. Because it produces ultrametric trees, this is generally not a good method. It is especially inappropriate for closely-related strains isolated at different times. Neighbor joining, minimum evolution, and least squares are a few methods that would be reasonable.

Answer to Reviewer:

That is a valid comment. We have done SNP analysis and created a Maximum Likelihood tree (S2 Fig) to serve as the reference analysis for the manuscript. A section has been added (lines 198-201 of the revised manuscript) describing the methods used. The parameters used and number of SNPs associated with the analysis are contained in S1 SNVPhyl analysis. As noted in lines 367-378 of the revised manuscript, the ML tree, the dendrogram in S3 Fig (revised manuscript), and the MSTs are extremely concordant – we actually did not expect to see the degree of concordance that was found. We have therefore kept the MST trees for the remaining analysis because it is much easier to interpret the data intuitively and at a glance when present in this form. We understand the sensitivity to calling MSTs phylogenetic trees and have therefore referred to them just as MSTs throughout the manuscript.

Note that we have replaced S3 Fig with a version that facilitates the clade comparison with the ML tree and also reflects changes resulting from new information on the categorization of phenotypically monophasic isolates from the S. Typhimurium designation given based on initial analysis back to 4,[5]:12:i:- based hin R140L and fljA A46T changes.

2. It is not clear to me whether the authors are suggesting multiple independent acquisitions within Clade IV. If so, why is this hypothesis favored over a single acquisition followed by different rearrangements? 

Answer to Reviewer: 

We suggest possible multiple independent acquisitions of the MREL and a single acquisition of SGI-4. Clonal descent of isolates containing these elements is also definitely occurring, and we have (re)added that concept in line 323-324 of the revised manuscript.

The strongest argument for multiple independent acquisitions is actually the presence of the MREL in a small number of isolates outside Clade IV (S4 Fig, revised manuscript), something that is highly unlikely to happen by clonal descent. These events appear to be unique and non-clonal in nature while the MREL is quite conserved, suggesting that some kind of mobilization of the MREL is possible. How much is difficult to say at this point. See lines380-384 of the revised manuscript.

It also depends on how one chooses to interpret the data on the lesions at the site of MREL insertion, including the placement of the MREL adjacent to iroB. We have added S4 Fig, which compares the different combinations of hin (presence or absence), fljA (presence or absence), and fljB (presence or absence, point mutation) with the presence of the MREL and SGI-4. It is easy to imagine how the genotype of PNCS015322 (no fljAB, hin is present) could have arisen by rearrangement from the closest isolates, PNCS015388 and PNCS015274 (no fljAB no hin). But it is also possible that this could have arisen through novel acquisition of the MREL in this strain, creating a different genotype. A little bit further down in the dendrogram we have a very tight cluster of isolates containing PNCS015513, PNCS014412, PNCS015529, PNCS015530 which have the genotypes fljA+ fljB(SNP1) no hin, fljA+ fljB(SNP1) + hin, etc. It’s a bit more difficult to imagine how the hin came into the cluster. I think the mechanism is not yet settled, and have tried to discuss it in this way in lines 390-403 of the revised manuscript.

We have tried to use the phrase “consistent with” in the manuscript to indicate that I thought that multiple insertions was possible but not proven. Unfortunately, the research that would prove one or the other possibility is not close enough to program objectives for me to follow up myself

3. I was initially confused by the use of "dcm". This is the name of a gene present in most Salmonella, which is different from the gene referred to here.

Answer to Reviewer: 

I agree. That is why I included the name of the protein product in the figures. Not enough short gene descriptions to go around, I guess.

4. Line 78: "the remaining iroB genes" There is only one iroB gene, right? Should this be "iro genes"? Or perhaps "iroB operon genes"?

Answer to Reviewer: 

Actually, there has been a partial deletion in some of the iroB genes. Line 81 of the revised manuscript has been revised to read, “…leaving an IS26 element adjacent to the remaining part of the iroB genes…” This is one of the cases where my interpretation is in favour of a fairly stable construction/lesion that may indicate a separate MREL insertion. There is usually a small gene between hin and iroB that was not annotated in many earlier works that has been removed in the case of these isolates as well as part of iroB.

My working hypothesis, by the way, is that whatever cointegrate is formed in the process of insertion of MREL between hin and iroB results in an imprecise excision of DNA, creating the different genetic lesions and deletions at the site. But I have no empirical evidence for it.

5. Line 92: "evade selection" is odd here. In fact it is selection that is presumably maintaining these elements. I imagine that the authors mean that selection against the element due to the cost of maintaining it is overcome by selection for resistance.

Answer to Reviewer: 

We have changed “evade” to “survive” on line 95 of the revised manuscript. 

6. Line 125. If these are on a plasmid in the Spanish clone, does that mean that they are not involved in the monophasic phenotype? If so, this distinction would be worth making.

Answer to Reviewer: 

The plasmids sequences and genome sequences do not appear to be available. This is what the reference said. I have found one instance where both the genome and plasmid of a single isolate contain all or part of the MREL, and think this may be the case here. But I don’t know. And I don’t know how complete the MREL is in the plasmid. It would be great if we could revisit the strains and figure it out, and the line in the manuscript was there in hopes someone would do that.

7. Line 138 "indels encoding heavy metal resistance". Strange way to put it. An insertion can encode resistance, but a deletion cannot.

Answer to Reviewer: 

The word was changed to “elements” (line 141, revised manuscript).

8. Line 149 "Isolate numbers were replaced with randomly generated numbers..." Why? Does Supporting spreadsheet 1 associate the published numbers with biosample IDs, or only the original numbers?

Answer to Reviewer: 

S1 Spreadsheet has been updated to provide the randomly generated numbers, the SRA accession numbers, and the NCBI biosample numbers. A couple of pages included by mistake were also removed. So people should be able to interpret data, get the raw reads from the sequence archive, and look at the project.

In Canada the provinces are responsible for health, which therefore resides with the provincial public health laboratories. They own the strains and all data associated with them and the federal laboratory can only do analysis with the prior permission of the provincial laboratories. Different provinces have different privacy laws, and it is difficult to satisfy all the requirements of all provinces. PulseNet Canada has chosen assignment of randomly generated numbers to replace isolate names as a means to make it more difficult for bad actors to identify patients from which the isolates were obtained. 

9. Line 243 Extra "within the"

Answer to Reviewer: 

Yep, that one would have been embarrassing. One of them has been removed. Thanks! Now line 250 of the revised manuscsript.

10. Line 327 I would not say that this information is "in addition to" phylogenetic analysis. Whether or not it is formally used that way, it is phylogenetic information that is distinct from the MLST genotypes.

Answer to Reviewer: 

Agreed. The wording in line 336 of the revised manuscript has been changed. Hopefully this is acceptable.

11. Fig. 7 "Complete genome". A more relevant, informative label would be "Chromosome".

Answer to Reviewer: 

We have been consistent throughout the manuscript to use “complete genome” to mean one that has been closed as opposed to a draft genome. We have both but used the complete genome as the reference.

12. I might state briefly what fljA, fljB, and hin are in the Introduction.

Answer to Reviewer: 

Yes, that is very important. I have added that to the beginning of the second paragraph of the Introduction, lines 54-57.

---

## [Decision Letter · Decision Letter 1]

8 Jul 2020

Distribution of heavy metal resistance elements in Canadian Salmonella 4,[5],12:i:- populations and association with the monophasic genotypes and phenotype

PONE-D-20-06506R1

Dear Dr. Clark,

We’re pleased to inform you that your manuscript has been judged scientifically suitable for publication and will be formally accepted for publication once it meets all outstanding technical requirements.

Kind regards,

Nicholas J Mantis

Academic Editor

PLOS ONE

Additional Editor Comments (optional):

Reviewers' comments:

Reviewer's Responses to Questions

**Comments to the Author**

1. If the authors have adequately addressed your comments raised in a previous round of review and you feel that this manuscript is now acceptable for publication, you may indicate that here to bypass the “Comments to the Author” section, enter your conflict of interest statement in the “Confidential to Editor” section, and submit your "Accept" recommendation.

Reviewer #1: All comments have been addressed

2. Is the manuscript technically sound, and do the data support the conclusions?

Reviewer #1: (No Response)

3. Has the statistical analysis been performed appropriately and rigorously? 

Reviewer #1: (No Response)

4. Have the authors made all data underlying the findings in their manuscript fully available?

Reviewer #1: (No Response)

5. Is the manuscript presented in an intelligible fashion and written in standard English?

Reviewer #1: (No Response)

6. Review Comments to the Author

Reviewer #1: (No Response)

7. PLOS authors have the option to publish the peer review history of their article (what does this mean?). If published, this will include your full peer review and any attached files.

Reviewer #1: No

---

## [Editor Report · Acceptance letter]

10 Jul 2020

PONE-D-20-06506R1 

Distribution of heavy metal resistance elements in Canadian *Salmonella* 4,[5],12:i:- populations and association with the monophasic genotypes and phenotype 

Dear Dr. Clark:

I'm pleased to inform you that your manuscript has been deemed suitable for publication in PLOS ONE. Congratulations! Your manuscript is now with our production department. 

Kind regards, 

on behalf of

Dr. Nicholas J Mantis 

Academic Editor

PLOS ONE